# S'MoRE: Structural Mixture of Residual Experts for Parameter-Efficient LLM Fine-tuning

**Hanqing Zeng**
Meta AI
zengh@meta.com

**Yinglong Xia**
Meta AI
yxia@meta.com

**Zhuokai Zhao**
Meta AI
zhuokai@meta.com

**Chuan Jiang**
Meta AI
gjiang@meta.com

**Qiang Zhang**
Meta AI
qiangzhang@meta.com

**Jiayi Liu**
Meta AI
liujiayi@meta.com

**Qunshu Zhang**
Meta AI
qunshuzhang@meta.com

**Lizhu Zhang**
Meta AI
lizhu@meta.com

**Xiangjun Fan**
Meta AI
maxfan@meta.com

**Benyu Zhang**
Meta AI
byzhang@meta.com

## Abstract

Fine-tuning pre-trained large language models (LLMs) presents a dual challenge of balancing parameter efficiency and model capacity. Existing methods like low-rank adaptations (LoRA) are efficient but lack flexibility, while Mixture-of-Experts (MoE) enhance model capacity at the cost of more & under-utilized parameters. To address these limitations, we propose **S**tructural **M**ixture **o**f **R**esidual **E**xperts (**S'MoRE**), a novel framework that seamlessly integrates the efficiency of LoRA with the flexibility of MoE. Conceptually, S'MoRE employs *hierarchical* low-rank decomposition of expert weights, yielding residuals of varying orders *interconnected* in a multi-layer structure. By routing input tokens through sub-trees of residuals, S'MoRE emulates the capacity of numerous experts by instantiating and assembling just a few low-rank matrices. We craft the inter-layer propagation of S'MoRE's residuals as a special type of Graph Neural Network (GNN), and prove that under similar parameter budget, S'MoRE improves *structural flexibility* of traditional MoE (or Mixture-of-LoRA) by exponential order. Comprehensive theoretical analysis and empirical results demonstrate that S'MoRE achieves superior fine-tuning performance, offering a transformative approach for efficient LLM adaptation. Our implementation is available at: https://github.com/ZimpleX/SMoRE-LLM.

## 1   Introduction

Large language models (LLMs) have achieved remarkable success across a wide range of tasks by leveraging extensive pretraining on vast datasets, which equips them with general-purpose knowledge [Achiam et al., 2023, Dubey et al., 2024, Team et al., 2024a, Liu et al., 2024a, Anthropic, 2024]. However, the versatility of pre-trained LLMs often falls short when applied to specialized tasks [Hadi et al., 2023, Ling et al., 2023, Chen et al., 2024]. Fine-tuning addresses this limitation by refining LLM's capabilities to focus on the nuances of different domains [Zhang et al., 2023, Wang et al., 2024a, 2025]. However, it introduces a fundamental tension between balancing parameter efficiency and the need for expanded model capacity to capture task-specific complexity [Wang et al., 2024b].

Low-Rank Adaptations (LoRA) [Hu et al., 2021, Mao et al., 2025] are parameter efficient but lack the capacity required by complex tasks. On the other hand, Mixture-of-Experts (MoE) [Lepikhin et al., 2020, Fedus et al., 2022b, Cai et al., 2024, Dai et al., 2024] architectures improve model

39th Conference on Neural Information Processing Systems (NeurIPS 2025).

capacity by enabling conditional computation where different tokens activate different experts. However, traditional MoEs are often less parameter- or data-efficient since the multiple experts need to separately learn their own parameters. Moreover, increasing the number of experts poses the challenge of balancing their utilization with low routing overhead.

Thus, to improve model capacity while maintaining parameter efficiency, we look into other scaling dimensions, such as *routing flexibility* unveiled by recent literature. Dai et al. [2024], Ludziejewski et al. [2024], He [2024] empirically show that under the same parameter budget, a large number of small (i.e., fine-grained) experts is more powerful than a small number of large experts. Ludziejewski et al. [2024] quantitatively derives the MoE scaling law with respect to experts' granularity. DeepSeek-MoE [Dai et al., 2024] elaborates the intuition: Consider an MoE system with 16 rank-128 experts. Under top-2 routing, each token has $\binom{16}{2} = 120$ ways to select its own expert combination. If we break down each rank-128 expert as 4 rank-32 experts (and correspondingly use top-8 routing to keep the same activated parameters), each token now has $\binom{4 \times 16}{8} = 4.4\text{B}$ routing choices.

Yet, "breaking down experts into finer granularity" may not be the most effective way of increasing routing flexibility. There are two potential limitations: 1) in the parameter-efficient fine-tuning (PEFT) scenario, each expert is already of a low rank, making it questionable if finer granularity is desirable; and 2) with more number of experts comes higher requirement on the router's capability & the learning algorithm – it may be challenging to maintain good expert utilization when the number of experts grows. These lead to our goal: *without* increasing the number of experts, we aim at emulating more routing choices by exploiting the *power of structure*. That is, instead of merely addressing the problem of "which experts to activate" (like most existing literature), we further ask: "how should the activated experts be connected"? To answer it, we first arrange experts into multiple layers. Then the router iterates through the layers and constructs a tree of activated experts, through which the input token propagates. The key observation is that, the same set of experts can interconnect in different ways to form exponentially many *non-isomorphic* tree structures, each yielding a distinct output. We thus formalize such intuition by extending the aforementioned routing flexibility into a new metric, **structural flexibility**, and theoretically quantify its exponential growth enabled by S'MoRE's design.

**Proposed work.** We propose **S**tructural **M**ixture **o**f **R**esidual **E**xperts (🍈 S'MoRE), a novel PEFT architecture that improves MoE's model flexibility & capacity by exploiting experts' structural relationship, while being as parameter-efficient as LoRA. We start from hierarchical residual decomposition of expert weights, where low-rank parameters of different orders form a multi-layer inter-connection network. We craft the model architecture so that when residuals aggregate and propagate across layers, they 1. remain low-rank to maintain overall efficiency, 2. can generate distinct embeddings for *all* non-isomorphic router-selected sub-trees, which theoretically guarantees high structural flexibility, and 3. are able to express the standard single-layer MoE model variants, which makes S'MoRE a strictly more powerful upgrade. To customize the expert structure for each token, we design a hierarchical router that efficiently and iteratively selects the children residuals when traversing down the selected ancestors. S'MoRE can be conceptually seen as a novel Graph Neural Network (GNN), where the "graph" emerges dynamically from the router's selection, and the S'MoRE layers can simulate the graph isomorphism test [Huang and Villar, 2021, Xu et al., 2019] to ensure high structural flexibility. Overall, S'MoRE achieves the benefits of both LoRA and fine-grained MoE, and addresses their limitations by exploiting experts' structure – S'MoRE emulates the capacity of *exponentially more* experts than physically instantiated, while keeping each residual low-rank. We extensively evaluate S'MoRE on 3 base models (LLaMA 3.2 1B, LLaMA 3 8B and Gemma 2 9B), 7 fine-tuning benchmarks, 3 types of router gates, and across different scales. S'MoRE consistently and significantly outperform state-of-the-art models and the 1-layer baselines in terms of both accuracy and parameter efficiency, validating the direction towards better PEFT adapters via structural mixture.

## 2 Background and Related Work

**Parameter efficient fine-tuning (PEFT).** Given a pre-trained model, PEFT trains a light-weight adapter whose number of trainable parameters is just a small fraction of the pre-trained weights. LoRA [Hu et al., 2021] and its variants [Liu et al., 2024b, Kopiczko et al., 2024] achieve good empirical performance by learning only a low-rank matrix as the adapter, where rank controls the efficiency-accuracy tradeoff. Despite the high parameter efficiency, their model capacities are limited.

**Mixture-of-Experts (MoE).** Mixture-of-Experts designs have been shown to boost LLM's model capacity [Dai et al., 2024, Fedus et al., 2022b, Puigcerver et al., 2024] due to their flexibility in

conditionally activating different sets of parameters for different input tokens. Recent works have tailored MoE for PEFT. MixLoRA [Li et al., 2024a] constructs the adapter as a set of LoRA experts where each token activates its own top-$k$ experts. Similarly, MoLE [Wu et al., 2024b] considers a flat layer of LoRA experts and implements a flexible branch selection scheme. To enhance the mixing flexibility, SMoRA [Zhao et al., 2025] decomposes LoRA into single-rank fine-grained experts, and MoSLoRA [Wu et al., 2024a] integrates a subspace fusing matrix in the low-rank space. MoV [Zadouri et al., 2024] and MoLORA [Zadouri et al., 2024] proposes MoE variants that mix $(IA)^3$ vectors [Liu et al., 2022] or LoRA weights for adapting attention modules. HydraLoRA [Tian et al., 2024] splits LoRA's up-projection matrix into multiple heads, and then performs weighted sum of each head's output by the gating weights. In the existing PEFT-MoE models, the expert-selection gates often follow classic designs, such as the noisy top-$k$ [Shazeer et al., 2017] or the switch-transformer [Fedus et al., 2022b] gates. With more experts, it is more challenging to ensure all experts are well utilized [Fedus et al., 2022a]. Such limitation to model scale-up can be addressed by experts' structural composition in S'MoRE, as we dramatically improve the structural flexibility (see definition in §3.5) without adding more experts. See Appendix A for other MoE variants.

## 2.1 Preliminaries

We consider one transformer layer and omit the layer index. Let $\boldsymbol{x} \in \mathbb{R}^d$ be the $d$-dimensional token embedding input to the adapter. The adapter's output $\boldsymbol{x}'$ is added back to the output generated by the pre-trained weights. All adapters here can be applied to weights of both FFN and attention modules.

**LoRA formulation.** The adapter consists of a down-projection matrix $\boldsymbol{A} \in \mathbb{R}^{d \times r}$ and an up-projection matrix $\boldsymbol{B} \in \mathbb{R}^{r \times d}$, with rank $r \ll d$ to achieve parameter efficiency. LoRA maps the input token embedding $\boldsymbol{x}$ to adapter's output $\boldsymbol{x}'$ as follows: $\boldsymbol{x}' = \boldsymbol{B} \cdot \boldsymbol{A} \cdot \boldsymbol{x}$.

**MoE formulation.** Let $s$ be the number of experts. The router maps each input token to different experts, where $\texttt{ROUTE}\,(\boldsymbol{x})^i \in \mathbb{R}$ gives expert $i$'s score for token $\boldsymbol{x}$. If the router performs top-$k$ sparse gating, then only top-$k$ values of $\texttt{ROUTE}\,(\boldsymbol{x})^i$ are kept and the rest are cast to 0. Suppose each expert $i$ performs linear transformation via matrix $\boldsymbol{W}^i$. The MoE layer performs:

$$\boldsymbol{x}' = \sum_{i=1}^{s} \texttt{ROUTE}\,(\boldsymbol{x})^i \cdot \boldsymbol{W}^i \cdot \boldsymbol{x} \tag{1}$$

# 3  🥟 S'MoRE

## 3.1 Low-Rank MoE Variants

**Mixture of low-rank experts (MoLRE).** To improve parameter efficiency of Eq. 1, we can approximate its $\boldsymbol{W}^i$ by some low rank $\boldsymbol{B}^i \cdot \boldsymbol{A}^i$ as defined in §2.1 (e.g., we can perform SVD on $\boldsymbol{W}^i$ and derive $\boldsymbol{B}^i \cdot \boldsymbol{A}^i$ corresponding to the largest singular values). We term such a model family as *mixture of low-rank experts (MoLRE)* [Wu et al., 2024b, Dou et al., 2024, Li et al., 2024a]. MoLRE's operation is derived by updating Eq. 1 as follows: $\boldsymbol{x}' = \sum_{i=1}^{s} \texttt{ROUTE}\,(\boldsymbol{x})^i \cdot \boldsymbol{B}^i \cdot \boldsymbol{A}^i \cdot \boldsymbol{x}$

**Mixture of multi-order residues (MoMOR).** We can generalize MoLRE's low-rank approximation into this form $\boldsymbol{W}^i \approx \sum_{\ell=0}^{L-1} \boldsymbol{B}_{\ell}^i \cdot \boldsymbol{A}_{\ell}^i$, where each $\boldsymbol{B}_{\ell}^i \cdot \boldsymbol{A}_{\ell}^i$ has a low rank (so MoLRE corresponds to $L = 0$). We call $\boldsymbol{B}_{\ell}^i \cdot \boldsymbol{A}_{\ell}^i$ as the $(\ell+1)^{\text{th}}$-order residual term, and denote its rank as $r_{\ell}$. The sum $\sum_{\ell=0}^{L-1} \boldsymbol{B}_{\ell}^i \cdot \boldsymbol{A}_{\ell}^i$ can have a rank up to $\sum_{\ell=0}^{L-1} r_{\ell}$, which is higher than the individual residuals.

We thus introduce *mixture of multi-order residues (MoMOR)*, an extension to MoLRE. Let $\mathcal{R}_{\ell} = \{\boldsymbol{B}_{\ell}^1 \cdot \boldsymbol{A}_{\ell}^1, \boldsymbol{B}_{\ell}^2 \cdot \boldsymbol{A}_{\ell}^2, \ldots\}$ be the set of order-$(\ell+1)$ residues, MoMOR model performs the following:

$$\boldsymbol{x}' = \sum_{\ell=0}^{L-1} \sum_{i=1}^{s_{\ell}} \texttt{ROUTE}_{\ell}\,(\boldsymbol{x})^i \cdot \boldsymbol{B}_{\ell}^i \cdot \boldsymbol{A}_{\ell}^i \cdot \boldsymbol{x} \tag{2}$$

where the model dynamically selects and combines different orders of residuals via routing. MoMOR can adaptively distribute computation across different levels of approximation, improving efficiency and expressivity. Notably, when we set $L = 2$ and $\texttt{ROUTE}_0\,(\boldsymbol{x})^i$ as a dense gate, the order-1 experts are activated for all tokens. MoMOR becomes a *shared-expert* MoE. This is a design adopted by DeepSeek-v3 [DeepSeek-AI, 2024] and many others [Rajbhandari et al., 2022, Li et al., 2024a].

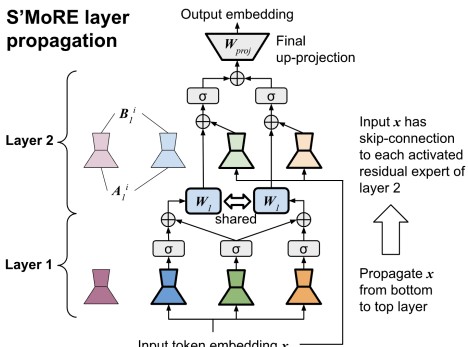
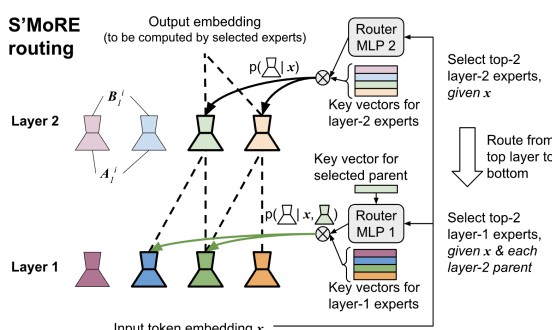

(a) Propagation of residuals across multiple S'MoRE layers (see Eq. 3). Here we consider 2 layers. Layer 1 has 3 activated residuals, where the dark green residual is selected by both the light green and the light orange parents in layer 2.

(b) Recursive routing of S'MoRE (§3.3). The router first selects the layer 2 residuals for token $\boldsymbol{x}$. Then it selects the layer 1 children conditioned on the activated layer 2 parent. We use a lightweight MLP to generate the query vector from the token embedding and the parent's key embedding.

Figure 1: Illustration of the layer propagation and routing process of S'MoRE.

## 3.2 Structural Mixture

In the following, "layer" refers to a S'MoRE layer with a collection of residual experts, rather than a transformer layer. Based on MoMOR, we arrange all the residues $\mathcal{R}_0, \ldots, \mathcal{R}_{L-1}$ into a $L$-layer structure. For each token $\boldsymbol{x}$, we activate a sub-structure that interconnects correlated residues in adjacent layers. The token propagates along the sub-structure layer by layer. Each layer implements a lightweight function to aggregate previous-layer residues. Extending the standard MoE to multiple layers improves model capacity by drastically increasing the model's *structural flexibility* (§3.5).

**Parameters.** Let $\boldsymbol{x} \in \mathbb{R}^d$ be the $d$-dimensional token embedding. Layer $\ell + 1$ (for $0 \leq \ell \leq L - 1$) consists of $s_\ell$ residual experts. Each expert $i$ (with $1 \leq i \leq s_\ell$) consists of a down-projection matrix $\boldsymbol{A}_\ell^i \in \mathbb{R}^{r_\ell \times d}$ and an up-projection matrix $\boldsymbol{B}_\ell^i \in \mathbb{R}^{d_{\ell+1} \times r_\ell}$, where $r_\ell$ is the experts' rank and $d_{\ell+1}$ is the output dimension of layer $\ell + 1$. Layer $\ell + 1$ also has a learnable $\boldsymbol{W}_\ell \in \mathbb{R}^{d_{\ell+1} \times d_\ell}$ that projects the layer-$\ell$ output to the $d_{\ell+1}$-dimensional subspace. Thus, $\mathcal{R}_\ell$ consists of all $\boldsymbol{A}_\ell^i$ and $\boldsymbol{B}_\ell^i$ for $1 \leq i \leq s_\ell$.

**Propagation.** Token $\boldsymbol{x}$ propagates in the $L$-layer structure in two phases. In the *routing* phase, the router activates the best-matching experts *top-down* (from layer $L$ to 1). At layer $L$, the router selects experts from $\mathcal{R}_{L-1}$ using standard gates (e.g., Fedus et al. [2022b]). At an intermediate layer $\ell < L$, the router computes the score to activate an expert in $\mathcal{R}_{\ell-1}$, *conditioned* on the already activated ancestors in layers $\ell' > \ell$. This ensures the selected children are connected to their activated parents. Different from the traditional routers, the S'MoRE router customizes a depth-$L$ "residual tree" for each token. See §3.3 for router architecture and §3.5 for structural flexibility of the tree-based routing.

In the *aggregation* phase, the token propagates along the activated residual tree *bottom-up* (from layer 1 to $L$). Layer $\ell + 1$ aggregates the information from the activated children experts in $\mathcal{R}_\ell$, and generates output embedding for the parent expert in $\mathcal{R}_{\ell+1}$. For each parent expert $i$, define $\mathcal{N}_\ell^i$ as the set containing the indices of $i$'s children experts[1]. Layer $\ell + 1$ operates as follows:

$$\boldsymbol{x}_{\ell+1}^i = \sum_{n \in \mathcal{N}_\ell^i} \alpha_\ell^{i,n} \cdot \sigma \left( \boldsymbol{B}_\ell^n \cdot \boldsymbol{A}_\ell^n \cdot \boldsymbol{x} + \boldsymbol{W}_\ell \cdot \boldsymbol{x}_\ell^n \right) \tag{3}$$

where $\sigma(\cdot)$ is a non-linear function which can be just an activation (e.g., ReLU [Agarap, 2018]). The scalar $\alpha_\ell^{i,n}$ is the router-generated score, elaborated in §3.3. Inputs to Eq. 3 consist of two parts: 1) Raw token embedding $\boldsymbol{x}$, which acts as skip connection to residuals $\boldsymbol{B}_\ell^n \cdot \boldsymbol{A}_\ell^n$ of various orders; and 2) $\boldsymbol{x}_\ell^n$ output from the previous layer, which enables *deep* interaction among multi-order residuals given non-linear $\sigma(\cdot)$ (compared to the shallow aggregation in Eq. 2). For $\ell = 0$, input $\boldsymbol{x}_0^n$ does not exist. To simplify notation, we define $d_0 := 0$, making $\boldsymbol{x}_0^i \in \mathbb{R}^0$ and $\boldsymbol{W}_0 \in \mathbb{R}^{d_1 \times 0}$ as an empty vector /

---

[1]We abuse notation here for ease of description. The nuance is that the same expert can be activated multiple times by different parents / ancestors. So $i$ should refer to the index of a node in the activated tree, rather than the index of just an expert. Similarly, superscript $n$ of $\boldsymbol{x}_\ell^n$ should be updated to $i \to n$ as a unique identifier (otherwise it creates ambiguity when expert $n$ is a child of multiple parents). See Eq. 32 in Appendix §3.4.

matrix. Then Eq. 3 applies to all layers $0 \leq \ell \leq L - 1$. The last layer $L$ has a single output node (i.e., $s_L = 1$) generated by aggregating information from the entire residual tree. Define $\boldsymbol{x}_L := \boldsymbol{x}_L^0$.

**Dimensionality** $d_\ell$. We should set the output $d_\ell$ to 1) avoid information loss in the aggregation process, and 2) keep the overall $L$-layer propagation efficient. A naïve choice following LoRA is $d_{\ell+1} = d \gg r_\ell$ (e.g., $d = 4096$, $r_\ell = 16$), which makes multiplication with $\boldsymbol{W}_\ell$ prohibitively expensive. To reduce cost, we should find the smallest $d_{\ell+1}$ that preserves the same amount of information as the vanilla setting $d_{\ell+1} = d$. The problem is equivalent to finding the maximum dimension of the subspace that $\boldsymbol{x}_{\ell+1}^i$ (Eq. 3) can span for any $\mathcal{N}_\ell^i, \boldsymbol{B}_\ell^n, \boldsymbol{A}_\ell^n$ and activated $i$. To simplify discussion, ignore activation $\sigma(\cdot)$: 1) For any $\boldsymbol{x}$, output $\boldsymbol{B}_\ell^n \cdot \boldsymbol{A}_\ell^n \cdot \boldsymbol{x}$ maximally spans a $d'$-dimensional subspace of the original $\mathbb{R}^d$, where $d' = \min\{d_{\ell+1}, r_\ell\}$; 2) There are $s_\ell$ possible $n$, leading to $s_\ell$ different $d'$-dimensional subspaces. When mutually orthogonal, they maximally span $\min\{d_{\ell+1}, s_\ell \cdot r_\ell\}$ dimensions; 3) $\boldsymbol{W}_\ell \cdot \boldsymbol{x}_\ell^n$ can span another subspace of dimension $\min\{d_{\ell+1}, d_\ell\}$ defined by $\boldsymbol{W}_\ell$ (independent of $n$). So $\boldsymbol{B}_\ell^n \cdot \boldsymbol{A}_\ell^n \cdot \boldsymbol{x} + \boldsymbol{W}_\ell \cdot \boldsymbol{x}_\ell^n$ maximally spans $d'' = \min\{d_{\ell+1}, d_\ell + s_\ell \cdot r_\ell\}$ dimensions; 4) Since a subspace is closed under linear combinations, $\sum_{n \in \mathcal{N}_\ell^i} \alpha_\ell^{i,n} (\boldsymbol{B}_\ell^n \cdot \boldsymbol{A}_\ell^n \cdot \boldsymbol{x} + \boldsymbol{W}_\ell \cdot \boldsymbol{x}_\ell^n)$ remains in the $d''$-dimensional subspace, regardless of $\mathcal{N}_\ell^i$. For the vanilla case $d_{\ell+1} = d$ with large enough $d$, we have $d'' = \min\{d_{\ell+1}, d_\ell + s_\ell \cdot r_\ell\} = d_\ell + s_\ell \cdot r_\ell$. Thus, the minimum $d_{\ell+1}$ is $d''$:

$$d_{\ell+1} = d_\ell + s_\ell \cdot r_\ell \qquad \Rightarrow \qquad d_\ell = \sum_{i=0}^{\ell-1} s_i \cdot r_i, \;\; \text{where } d_0 := 0 \text{ and } \ell \in [0, L-1] \qquad (4)$$

**Final projection.** After the last layer $L$, we map the $d_L$-dimensional output $\boldsymbol{x}_L$ to the final output dimension $d_{\text{out}}$ (i.e., $d_{\text{out}}$ is the dimensionality of $\boldsymbol{x}'$ in Eq. 1 and Eq. 2). We thus have a projection matrix $\boldsymbol{W}_{\text{proj}} \in \mathbb{R}^{d_L \times d}$ that simply performs $\boldsymbol{x}' = \boldsymbol{W}_{\text{proj}} \cdot \boldsymbol{x}_L$.

### 3.3 Hierarchical Routing

Fig. 1b illustrates the top-down routing. We start from layer $L$. The router computes $p(i_{L-1} \mid \boldsymbol{x})$, the probability to activate an expert $i_{L-1}$ in $\mathcal{R}_{L-1}$ given token $\boldsymbol{x}$. The top-$f_{L-1}$ experts with the highest $p(i_{L-1} \mid \boldsymbol{x})$ are selected. Next, for each selected expert $i_{L-1}$, we compute $p(i_{L-2} \mid i_{L-1}, \boldsymbol{x})$, which is the conditional probability to activate $i_{L-2}$ in $\mathcal{R}_{L-2}$ given its activated parent $i_{L-1}$ and $\boldsymbol{x}$. Each activated $i_{L-1}$ further activates $f_{L-2}$ children with the highest $p(i_{L-2} \mid i_{L-1}, \boldsymbol{x})$. Generally, the router computes the conditional probability $p(i_{\ell-1} \mid i_{L-1}, \ldots i_\ell, \boldsymbol{x})$, with $i_{L-1}, \ldots i_\ell$ being all the activated ancestors of the candidate $i_{\ell-1}$. All activated experts form a depth-$L$ tree. Each depth-$\ell$ node fans out to $f_{L-\ell-1}$ children experts (the activated layer-$L$ experts are the depth-1 tree nodes).

Let $f_\ell$ be the fanout factor of each parent expert, $F_\ell$ be the total number of experts selected from $\mathcal{R}_\ell$ (i.e., $F_\ell$ is the total number of depth-$(L-\ell)$ experts in the activated tree). The same expert can be selected multiple times by ancestors on different paths – It is possible that $F_\ell > s_\ell$. We derive $F_\ell$ as:

$$F_\ell = \prod_{i=\ell}^{L-1} f_i \qquad (5)$$

**Router architecture.** For each expert $i$ in $\mathcal{R}_\ell$, we instantiate a learnable $m$-dimensional *key* vector $\boldsymbol{k}_\ell^i \in \mathbb{R}^m$. For the whole candidate pool $\mathcal{R}_\ell$, we instantiate a neural network, $\texttt{MLP}_\ell(\cdot)$, to generate an $m$-dimensional *query* vector based on $\boldsymbol{x}$ and the ancestors. The routing probability over $\mathcal{R}_\ell$ is computed by the normalized key-query dot product. For a path of activated ancestors, "expert $i'$ in $\mathcal{R}_{\ell+1}$, ..., expert $i'^{\cdots'}$ in $\mathcal{R}_{L-1}$", the router generates the query vector $\boldsymbol{q}$ and the router score $\alpha_\ell^i$ as follows, where $\texttt{concat}(\cdot)$ performs vector concatenation and $\texttt{softmax}(\cdot)$ normalizes over $\mathcal{R}_\ell$.

$$\boldsymbol{q} = \texttt{MLP}_\ell\left(\texttt{concat}\left(\boldsymbol{x}, \boldsymbol{k}_{\ell+1}^{i'}, \cdots, \boldsymbol{k}_{L-1}^{i'^{\cdots'}}\right)\right) \qquad (6)$$

$$\alpha_\ell^i = \texttt{softmax}\left(\langle \boldsymbol{k}_\ell^i, \boldsymbol{q} \rangle\right) \qquad (7)$$

**Computation optimization.** Eq. 6 can be computationally expensive when all $\texttt{MLP}_\ell(\cdot)$ need to process the high-dimensional $\boldsymbol{x}$. To reduce computation, we first project the $d$-dimensional $\boldsymbol{x}$ to a $d_{\text{down}}$-dimensional $\boldsymbol{x}_{\text{down}}$ (e.g., $d = 4096$, $d_{\text{down}} = 24$), and then replace $\boldsymbol{x}$ with $\boldsymbol{x}_{\text{down}}$ in Eq. 6. The dimension of the input to $\texttt{MLP}_\ell(\cdot)$ then becomes $d_{\text{down}} + (L - \ell - 1) \cdot m$.

**Gating types.** Our router and layer designs are compatible with various types of gates. In our experiments (§4), we have evaluated: 1. *Dense* gate [Tian et al., 2024], which activates all children

experts ($f_\ell = s_\ell$); 2. *Sparse* noisy top-$k$ gate [Shazeer et al., 2017]; 3. *Sparse* switch gate [Fedus et al., 2022b]. The two sparse gates only activate a subset of the children experts ($f_\ell < s_\ell$) by the top routing scores $\alpha$. To avoid expert under-utilization and ensure all experts see sufficient amount of tokens during training, we implement an auxiliary load-balance loss according to the original papers [Shazeer et al., 2017, Fedus et al., 2022b]. See Appendix B.1 for more algorithmic details.

## 3.4 Parameter & Computation Efficiency

Although S'MoRE introduces structural learning modules, our design ensures **similar efficiency to the vanilla LoRA** (w.r.t. both computation and trainable parameters) under the same total rank.

**Parameter efficiency.** Each S'MoRE layer $\ell + 1$ consists of the following trainable parameters: $\boldsymbol{B}_\ell^n$, $\boldsymbol{A}_\ell^n$ and $\boldsymbol{W}_\ell$. The total trainable parameters equals:

$$P_{\ell+1} = s_\ell \cdot (d \cdot r_\ell + r_\ell \cdot d_{\ell+1}) + d_\ell \cdot d_{\ell+1} = s_\ell \cdot d \cdot r_\ell + d_{\ell+1} \cdot (s_\ell \cdot r_\ell + d_\ell) \stackrel{\text{(a)}}{=} s_\ell \cdot d \cdot r_\ell + d_{\ell+1}^2 \quad (8)$$

where the last step "(a)" is according to Eq. 4. The final projection matrix (end of §3.2) requires $P_{\text{proj}} = d \cdot d_L$ parameters. So the total number of parameters for all S'MoRE layers equals:

$$P_{\text{proj}} + \sum_{\ell=1}^{L} P_\ell = d \cdot d_L + d \cdot \left( \sum_{\ell=0}^{L-1} s_\ell \cdot r_\ell \right) + \Delta \stackrel{\text{(b)}}{=} 2 \cdot d \cdot d_L + \Delta \stackrel{\text{(c)}}{\approx} 2 \cdot d \cdot d_L \quad (9)$$

where $\Delta = \sum_{\ell=1}^{L} d_\ell^2$. Step "(b)" is by Eq. 4; $\Delta$ is the overhead due to multi-layer propagation. Since $d_1 < \ldots < d_L \ll d$ (e.g., $d_L = 64$, $d = 4096$), we have $\Delta \ll 2 \cdot d \cdot d_L$. This justifies step "(c)". In Table 1, we empirically validated the small overhead $\Delta$. With $f_\ell = 2$, $s_\ell = 4$, and $r_\ell = 8$ or $16$ for all layers $\ell$ (consistent with the §4 experiments), $\Delta$ is no more than **2%** for 2-layer S'MoRE.

The router's trainable parameters come from: 1) down-projection for $\boldsymbol{x}_{\text{down}}$, which requires $d \cdot d_{\text{down}}$ parameters, 2) per-layer "query" MLP. By §3.3, the MLP's input dimension is $d_{\text{down}} + (L - \ell) \cdot m$, where $m \ll d$ is the dimension of the "key" vectors. In practice, we set the MLP hidden dimension as $m$. Since $m$ and $d_{\text{down}}$ are both very small, the router's parameter count is practically negligible.

Table 1: Overhead $\Delta$ compared with the main computation cost $2 \cdot d \cdot d_L$

| $r_\ell$ | $L$ | $d_L$ | $2 \cdot d \cdot d_L$ | $\Delta$ | Overhead ratio |
|---|---|---|---|---|---|
| | 2 | 64 | 0.5M | 0.005M | 1.0% |
| 8 | 3 | 96 | 0.8M | 0.014M | 1.8% |
| | 4 | 128 | 1.0M | 0.031M | 2.9% |
| | 2 | 128 | 1.0M | 0.020M | 2.0% |
| 16 | 3 | 192 | 1.6M | 0.057M | 3.6% |
| | 4 | 256 | 2.1M | 0.123M | 5.9% |

In total, S'MoRE approximately has $2 \cdot d \cdot d_L$ parameters – *the same as the parameter count for a vanilla LoRA* with rank $d_L$ (the 2 factor is due to LoRA's down- and up-project matrices $\boldsymbol{A}$ and $\boldsymbol{B}$).

**Computation cost.** Following similar steps, we can derive the overhead in computation. The computation cost of the baseline LoRA is $2 \cdot d \cdot d_L$. The overhead introduced by S'MoRE is $\Delta' \leq \sum_{\ell=0}^{L-1} F_\ell \cdot d_{\ell+1} \cdot (d_\ell + r_\ell)$, which is again *neglible* in practice. See Appendix C.1 for details.

## 3.5 Model Capacity

We theoretically show S'MoRE enhances model capacity compared with baselines (see Appendix C.2 for proofs). First, we show that the two low-rank MoE variants in §3.1 are special cases of S'MoRE.

**Proposition 3.1.** *S'MoRE can express MoLRE, when $L = 1$ and $\sigma(\cdot)$ is the identity mapping.*

**Proposition 3.2.** *S'MoRE can express MoMOR, when setting $\sigma(\cdot)$ as the identity mapping.*

For any MoLRE (or MoMOR) model, we can find a corresponding S'MoRE that generates identical output as MoLRE (or MoMOR) for any input $\boldsymbol{x}$. Without $\sigma$, we can collapse a multi-layer S'MoRE into a single layer equivalent, where the dimensionality set by Eq. 4 ensures the same rank as MoMOR. Can S'MoRE be theoretically better than MoLRE and MoMOR, if we go beyond the constraints of Propositions 3.1 and 3.2 by setting $L > 1$ and $\sigma$ as non-linear mapping? To answer it, we analyze an MoE model's expressive power by quantifying the **structural flexibility**.

**Structural flexibility.** Let $\Theta$ be the collection of all experts' parameters ($\boldsymbol{B}_\ell^i$, $\boldsymbol{A}_\ell^i$ and $\boldsymbol{W}_\ell$ for $0 \leq \ell \leq L - 1$ and all $i$). Given $\Theta$, when a token $\boldsymbol{x}$ comes, different routers may activate different residual experts, and thus generate different output embedding $\boldsymbol{x}_L$. Therefore, we define $\texttt{dist}(\boldsymbol{x}; \Theta)$ as the number of *distinct* $\boldsymbol{x}_L$. The larger $\texttt{dist}(\boldsymbol{x}; \Theta)$ can be, the more "structurally flexible" the model architecture is. Our focus here is on the multi-layer structure formed by the residual experts, rather than the router network (thus, we assume an ideal router for the following Theorems).

Next we prove S'MoRE's higher model capacity by quantifying structural flexibility. In the following, we treat $\alpha_\ell^n$ as binary mask (1 for selected experts, and 0 otherwise) when generating $\boldsymbol{x}_L$.

**Theorem 3.3.** *The structural flexibility of MoMOR is upper-bounded by* $\Gamma_{\text{MoMOR}} = \max_{\boldsymbol{x},\Theta} dist\,(\boldsymbol{x};\Theta) \leq \binom{s_{L-1}}{f_{L-1}} \cdot \prod_{\ell=0}^{L-2}\left(\sum_{i=f_\ell}^{\min\{F_\ell,s_\ell\}}\binom{s_\ell}{i}\right).$

**Theorem 3.4.** *Setting* $\sigma\,(\cdot)$ *as an MLP, there exists some* $\Theta'$ *such that the structural flexibility of* S'MoRE *is:* $\Gamma_{\text{S'MoRE}} = \min_{\boldsymbol{x}} dist\,(\boldsymbol{x};\Theta') = \prod_{\ell=0}^{L-1}\binom{s_\ell}{f_\ell}^{F_{\ell+1}}$, *where we define* $F_L := 1$.

Above, $\binom{s}{k} = \frac{s!}{k!(s-k)!}$ is the binomial coefficient that quantifies the number of ways to choose $k$ out of $s$ items, ignoring order. $F_\ell$ is defined in Eq. 5. When increasing the number of layers, $\Gamma_{\text{S'MoRE}}$ exceeds the upper bound $\Gamma_{\text{MoMOR}}$ by orders of magnitude. The reason is that for MoMOR, the $\binom{s_\ell}{i}$ terms are summed over $F_\ell$, while for S'MoRE, $F_\ell$ becomes the **exponent** of $\binom{s_\ell}{f_\ell}$. In Fig. 2, we calculate the theoretical $\Gamma_{\text{MoMOR}}$ and $\Gamma_{\text{S'MoRE}}$ under depth $L$. Consistent with our experimental setup (§4.1), we set $s_\ell = 4$ and $f_\ell = 2$ for all $\ell$. Clearly, $\Gamma_{\text{S'MoRE}}$ is substantially higher than $\Gamma_{\text{MoMOR}}$ even for shallow models ($L = 2$), and $\Gamma_{\text{S'MoRE}}$ grows exponentially faster than $\Gamma_{\text{MoMOR}}$ when increasing $L$.

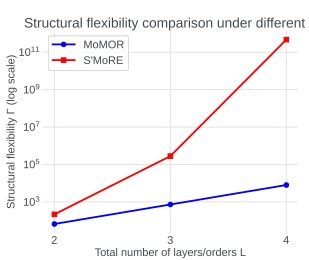

Figure 2: $\Gamma_{\text{S'MoRE}}$ and $\Gamma_{\text{MoMOR}}$ w.r.t. $L$ (with $s_\ell = 4$, $f_\ell = 2$).

We explain the intuition of the proof, and defer the details to Appendix C.2. *First*, $\Gamma_{\text{S'MoRE}}$ quantifies the number of non-isomorphic depth-$L$ trees that can be formed by any router. Each node at tree-level $\ell$ (i.e., an expert in $\mathcal{R}_{L-\ell}$; the same expert may appear multiple times at tree-level $\ell$ under different ancestor paths) has $\omega = \binom{s_{L-\ell-1}}{f_{L-\ell-1}}$ ways of selecting its children set. All nodes at tree-level $\ell$ jointly contribute to a $\omega^{F_{L-\ell}}$ factor. *Secondly*, S'MoRE can generate distinct outputs for all non-isomorphic sub-trees. We borrow conclusions from the Graph Neural Network literature. We view Eq. 3 as defining a variant of Graph Isomorphism Network (GIN) [Xu et al., 2019]. S'MoRE's $L$-layer propagation

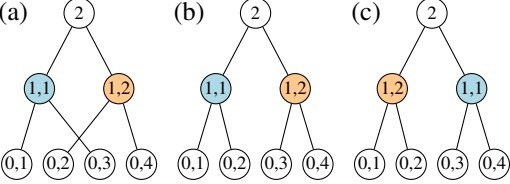

Figure 3: Examples where the same set of activated experts interconnect differently. MoMOR always generates the same output for (a), (b) and (c), while S'MoRE can distinguish all the three cases. A variant of S'MoRE that performs activation $\sigma$ differently (§3.6) can differentiate (a) from (b) or (c), but cannot differentiate (b) from (c). Note that (b) and (c) differ by swapped "1,1" and "1,2".

simulates the $L$-iteration Weisfeiler-Lehman (WL) test [Huang and Villar, 2021], where including non-linearly activated $\sigma$ is the key to ensure an injective "color refinement" process in WL. It then follows that the $L$ layer S'MoRE can distinguish non-isomorphic trees of depth $L$. *Third*, without activation $\sigma$, S'MoRE degrades to MoMOR, and is unable to distinguish many non-isomorphic depth-$L$ trees. Fig. 3 shows 3 examples with $L = 2$. Node 2 is the final output node (tree root). When we activate the same set of experts ("0,1", "0,2", "0,3", "0,4", "1,1", "1,2") but connect them differently (non-isomorphic), MoMOR always generates the same output while S'MoRE can produce different ones. This shows $\Gamma_{\text{MoMOR}} < \Gamma_{\text{S'MoRE}}$ and S'MoRE's higher expressivity.

### 3.6 Model Variants

**How activation $\sigma$ affects structural learning.** Theorem 3.4 concretely shows the benefit of including activation $\sigma$ in Eq. 3. What if we tweak Eq. 3 to let $\sigma$ operate on $\boldsymbol{x}_\ell^n$ rather than $\boldsymbol{B}_\ell^n \cdot \boldsymbol{A}_\ell^n \cdot \boldsymbol{x} + \boldsymbol{W}_\ell \cdot \boldsymbol{x}_\ell^n$?

$$\boldsymbol{x}_{\ell+1}^i = \sum_{n\in\mathcal{N}_\ell^i} \alpha_\ell^{i,n} \cdot (\boldsymbol{B}_\ell^n \cdot \boldsymbol{A}_\ell^n \cdot \boldsymbol{x} + \boldsymbol{W}_\ell \cdot \sigma\,(\boldsymbol{x}_\ell^n)) \tag{10}$$

We can then decompose Eq. 38 as $\sum_{n\in\mathcal{N}_\ell^i}\boldsymbol{B}_\ell^n\cdot\boldsymbol{A}_\ell^n\cdot\boldsymbol{x} + \sum_{n\in\mathcal{N}_\ell^i}\boldsymbol{W}_\ell\cdot\sigma\,(\boldsymbol{x}_\ell^n)$ (ignoring $\alpha_\ell^{i,n}$ for simplicity), and use Fig. 3 as an example to understand its expressive power. Trees (a) and (b) have the same layer-2 experts, "1,1" and "1,2", making their gray terms equivalent. Yet, their different layer-1 children combinations (tree (a) has "0,1" + "0,3" and "0,2" + "0,4", while tree (b) has "0,1" + "0,2" and "0,3" + "0,4") make their green terms different. This enables Eq. 38 to differentiate (a) from (b). Following this reasoning, for (b) and (c), their gray and green terms are both equal. Thus, Eq. 38 yields identical outputs for the two trees, even though they are non-isomorphic.

**Corollary 3.5.** *Let $\Gamma^\ell_{S'MoRE *}$ be the structural flexibility of $\ell$-layer $S'MoRE$ variant under Eq. 38. It satisfies the following recursion:* $\Gamma^\ell_{S'MoRE *} = \binom{s_{\ell-1}}{f_{\ell-1}} \cdot \binom{\Gamma^{\ell-1}_{S'MoRE *} + f_{\ell-1} - 1}{f_{\ell-1}}$, *where* $\Gamma^0_{S'MoRE *} := 1$.

It is easy to see that S'MoRE under Eq. 3 is more expressive than S'MoRE * under Eq. 38. Further, both S'MoRE variants are stronger than the baseline 1-layer MoEs. This is also illustrated by Fig. 3.

**S'MoRE with cross-layer parameter sharing.** We introduce S'MoRE #, another useful variant which lets the experts of different layers share the same parameters. i.e., $s := s_\ell$ and $r := r_\ell$ are the same for all layers $\ell$. And $\boldsymbol{A}^i := \boldsymbol{A}^i_\ell$ and $\boldsymbol{B}^i := \boldsymbol{B}^i_\ell$ for all $\ell$ and $1 \leq i \leq s$. This means experts in different layers now operate in the same embedding subspace, and hence the intermediate hidden dimension $d := d_\ell$ is the same for all $\ell$ – we update Eq. 4 as $d = s \cdot r$. The layers still propagate by Eq. 3.

We summarize the properties of S'MoRE #. Following similar derivation[2] in §3.4 (plugging in $d$ above), we conclude S'MoRE # has comparable parameter & computation efficiency as the vanilla LoRA. The structural flexibility below also has a similar form as Theorem 3.4 – S'MoRE and S'MoRE # exponentially boost structural flexibility of the 1-layer baselines, MoMOR and MoLRE, respectively.

**Corollary 3.6.** *The structural flexibility of $S'MoRE^\#$ equals* $\prod_{\ell=0}^{L-1} \binom{s}{f}^{F_{\ell+1}}$ *where* $F_L := 1$.

**Alternative router design (bottom-up version).** In addition to the top-down router in §3.3, we can also perform bottom-up routing, making the routing and layer propagation flow along the same direction. The bottom-up router still aims at customizing different children experts for different parents. Yet, when routing bottom-up, the parent index is unknown when we select the children. So now the key vector $\boldsymbol{k}$ (see Eq. 6) is not directly associated with any specific parent expert. It instead represents a node position in the routing tree. See Appendix B.2 for details and tradeoff discussion.

## 4 Experiments

### 4.1 Experimental Setup

**Datasets.** We fine-tune on a diverse set of benchmarks, including ARC-c/e [Clark et al., 2018], Commonsense QA (CSQA) [Talmor et al., 2018], OpenBook QA (OBQA) [Mihaylov et al., 2018], Winogrande [Sakaguchi et al., 2021], GSM8K [Cobbe et al., 2021], and HumanEval [Chen et al., 2021]. For HumanEval, we follow Tian et al. [2024] to train the base LLM on CodeAlpaca [Chaudhary, 2023], and evaluate "Pass@1" on HumanEval. For all other datasets, we fine-tune on the training split and evaluate "Accuracy" on the test split. See Appendix §D.1 for more details.

**Base models & baselines.** We use LLaMA 3.2-1B, LLaMA 3-8B [Dubey et al., 2024] and Gemma 2-9B [Team et al., 2024b] as the base models. We insert adapters of different kinds: 1. LoRA [Hu et al., 2021]; 2. mixture of LoRA experts (MixLoRA [Li et al., 2024a]): the state-of-the art parameter efficient MoE adapter, which is essentially the single-layer version of S'MoRE; 3. HydraLoRA [Tian et al., 2024]: another state-of-the-art PEFT adapter implementing a MoE variant of LoRA by splitting LoRA's up-projection $\boldsymbol{B}$ into multiple heads, and combining the multi-head outputs via scores from a dense gate; and 4. S'MoRE: the multi-layer extension of the above. To further evaluate the generalizability, we implement 3 variants of MixLoRA and S'MoRE using different gates (see §3.3 and Appendix B.1): 2 sparse gates (noisy top-$k$ [Shazeer et al., 2017] and switch-transformer [Fedus et al., 2022b] gates), and 1 dense gate (same as HydraLoRA [Tian et al., 2024]). See Appendix D.2.

**Training & evaluation methodology.** For hyperparameter tuning, we train all models using the same number of epochs, learning rate schedule, gradient accumulation steps and machine type. All models are trained under the LLaMA-Factory [Zheng et al., 2024] framework and evaluated with OpenCompass [Contributors, 2023b]. For hyperparameter search, we set an equal budget of trainable parameters, and vary the expert rank, the number of experts, the number of activated experts, etc. See Appendix D.2 for details of the hyperparameter range, and the hardware / software configuration.

### 4.2 Main Results

Table 2 presents the comprehensive comparison on accuracy and parameter efficiency. For all the base model and the gate type, we consistently observe that S'MoRE achieves **significant accuracy improvement without sacrificing parameter efficiency**. Specifically, 1. Among all the methods, while LoRA's parameter counts are low, its average accuracy is also the lowest. This implies the

---

[2]In S'MoRE #, the same expert may be activated in multiple layers. To avoid redundancy, we first collect the set of activated experts across all layers, and then compute $\boldsymbol{B}^i \cdot \boldsymbol{A}^i \cdot \boldsymbol{x}$ only once for each activated expert $i$.

Table 2: Comparison under two base models & three gate types. The hyperparameter search sets the same parameter budget for all models. The "Param." column denotes the trainable parameters (B) for the highest-accuracy model. In the "Method" column, number in parentheses denote the number of experts / heads ("4-4" denotes a 2-layer S'MoRE, each with 4 experts). Highest accuracy under the same gate is highlighted in **bold**, and highest accuracy across all gates is highlighted in red.

| | Gate | Method | ARC-c Acc. | ARC-c Param. | ARC-e Acc. | ARC-e Param. | CSQA Acc. | CSQA Param. | OBQA Acc. | OBQA Param. | Winogrande Acc. | Winogrande Param. | Avg Acc. | Avg Param. |
|---|---|---|---|---|---|---|---|---|---|---|---|---|---|---|
| **LLaMA 3.2 1B** | | Base | 32.54 | 0 | 66.31 | 0 | 23.67 | 0 | 43.80 | 0 | 50.75 | 0 | 43.41 | 0 |
| | | LoRA | 36.27 | 0.004 | 74.78 | 0.002 | 63.80 | 0.063 | 71.20 | 0.031 | 50.59 | 0.008 | 59.15 | 0.022 |
| | Dense | HydraLoRA (4) | 35.93 | 0.006 | 73.54 | 0.023 | 66.34 | 0.002 | 71.60 | 0.023 | 50.75 | 0.012 | 59.63 | 0.013 |
| | | HydraLoRA (8) | 35.93 | 0.012 | 72.31 | 0.007 | 62.08 | 0.042 | 71.60 | 0.012 | 50.99 | 0.012 | 58.58 | 0.017 |
| | | MixLoRA (4) | 39.66 | 0.021 | 72.84 | 0.134 | 65.44 | 0.134 | 70.40 | 0.134 | 51.30 | 0.007 | 59.93 | 0.086 |
| | | MixLoRA (8) | 39.32 | 0.021 | 74.78 | 0.270 | 66.42 | 0.069 | 69.60 | 0.134 | 51.14 | 0.037 | 60.25 | 0.106 |
| | | S'MoRE (2-2) | **40.00** | 0.017 | **75.31** | 0.085 | 66.99 | 0.037 | 72.20 | 0.085 | 52.01 | 0.015 | **61.30** | 0.048 |
| | | S'MoRE (4-4) | 39.66 | 0.017 | 74.43 | 0.085 | **67.32** | 0.045 | **72.80** | 0.202 | 52.01 | 0.168 | 61.24 | 0.103 |
| | Noisy top-$k$ | MixLoRA (4) | 39.32 | 0.037 | 71.96 | 0.069 | 64.70 | 0.134 | 70.00 | 0.134 | 51.46 | 0.069 | 59.49 | 0.089 |
| | | MixLoRA (8) | 37.97 | 0.069 | 72.84 | 0.270 | 65.03 | 0.134 | 70.80 | 0.270 | 51.46 | 0.069 | 59.62 | 0.162 |
| | | S'MoRE (2-2) | **39.66** | 0.029 | 73.19 | 0.135 | 64.95 | 0.135 | 70.00 | 0.102 | 51.54 | 0.029 | 59.87 | 0.086 |
| | | S'MoRE (4-4) | **39.66** | 0.037 | **74.96** | 0.135 | **66.26** | 0.102 | **71.40** | 0.135 | **52.17** | 0.273 | **60.89** | 0.136 |
| | Switch | MixLoRA (4) | 38.98 | 0.021 | 73.37 | 0.134 | 66.42 | 0.069 | 72.00 | 0.134 | 51.22 | 0.009 | 60.40 | 0.073 |
| | | MixLoRA (8) | 39.32 | 0.021 | 73.72 | 0.069 | 65.85 | 0.134 | 71.80 | 0.134 | 51.30 | 0.021 | 60.40 | 0.076 |
| | | S'MoRE (2-2) | 39.66 | 0.029 | 74.78 | 0.135 | 66.75 | 0.069 | 71.40 | 0.102 | **52.25** | 0.045 | 60.97 | 0.076 |
| | | S'MoRE (4-4) | **40.34** | 0.021 | 74.78 | 0.168 | **67.16** | 0.202 | 72.40 | 0.085 | 52.09 | 0.021 | **61.35** | 0.099 |
| **LLaMA 3 8B** | | Base | 80.34 | 0 | 89.77 | 0 | 70.35 | 0 | 73.80 | 0 | 59.91 | 0 | 74.83 | 0 |
| | | LoRA | 81.69 | 0.028 | 91.36 | 0.028 | 81.00 | 0.028 | 87.00 | 0.028 | 81.77 | 0.028 | 84.56 | 0.028 |
| | Dense | HydraLoRA (4) | **83.39** | 0.013 | 91.53 | 0.160 | 81.82 | 0.013 | 88.20 | 0.082 | 83.82 | 0.160 | 85.75 | 0.086 |
| | | HydraLoRA (8) | 81.69 | 0.079 | 91.53 | 0.015 | 81.49 | 0.024 | 86.60 | 0.015 | 84.14 | 0.297 | 85.09 | 0.086 |
| | | MixLoRA (4) | 81.69 | 0.026 | **92.24** | 0.247 | 81.24 | 0.033 | 89.40 | 0.478 | 84.06 | 0.247 | 85.73 | 0.206 |
| | | MixLoRA (8) | 82.37 | 0.132 | 91.71 | 0.247 | 81.00 | 0.033 | 88.60 | 0.075 | 85.40 | 0.478 | 85.82 | 0.193 |
| | | S'MoRE (2-2) | 82.37 | 0.090 | **92.24** | 0.190 | **81.90** | 0.037 | 89.40 | 0.054 | **88.24** | 0.480 | **86.83** | 0.170 |
| | | S'MoRE (4-4) | 82.71 | 0.190 | 91.89 | 0.247 | **81.90** | 0.033 | **90.00** | 0.076 | 85.48 | 0.247 | 86.40 | 0.157 |
| | Noisy top-$k$ | MixLoRA (4) | 82.37 | 0.075 | 91.53 | 0.247 | 80.75 | 0.075 | 87.80 | 0.075 | 82.00 | 0.478 | 84.89 | 0.190 |
| | | MixLoRA (8) | **83.39** | 0.950 | 91.53 | 0.247 | 80.67 | 0.075 | 88.40 | 0.247 | 83.19 | 0.478 | 85.44 | 0.399 |
| | | S'MoRE (2-2) | 82.37 | 0.305 | 91.36 | 0.090 | 81.82 | 0.104 | 88.20 | 0.047 | 83.27 | 0.190 | 85.40 | 0.147 |
| | | S'MoRE (4-4) | 82.37 | 0.104 | **91.71** | 0.305 | **82.06** | 0.047 | **90.00** | 0.480 | **85.48** | 0.714 | **86.32** | 0.330 |
| | Switch | MixLoRA (4) | 82.37 | 0.132 | **92.95** | 0.478 | 81.08 | 0.047 | 88.80 | 0.478 | 84.53 | 0.247 | 85.95 | 0.276 |
| | | MixLoRA (8) | 82.03 | 0.033 | 91.71 | 0.132 | 81.24 | 0.047 | 88.60 | 0.247 | 85.95 | 0.950 | 85.91 | 0.282 |
| | | S'MoRE (2-2) | 83.05 | 0.133 | 92.24 | 0.061 | 81.82 | 0.029 | **89.80** | 0.076 | **86.42** | 0.247 | 86.67 | 0.109 |
| | | S'MoRE (4-4) | **83.39** | 0.076 | 92.42 | 0.305 | **82.15** | 0.047 | **89.80** | 0.305 | 85.87 | 0.305 | **86.73** | 0.208 |

necessity of more advanced PEFT adapters of higher model capacity. 2. For models using dense gates, HydraLoRA achieves the lowest parameter count. However, its average accuracy is notably lower than both the 1-layer MoE model MixLoRA and the 2-layer S'MoRE. Since for all models, we set the same parameter budget for hyperparameter tuning, this means that HydraLoRA cannot effectively utilize more parameters to boost its accuracy (see also Fig. 4). 3. On all gate types, S'MoRE achieves significantly higher average accuracy than all baselines. In particular, MixLoRA belongs to the MoLRE family (§3.1) whose layer operation can be categorized by Eq. 1. Thus, it can be seen as a single-layer S'MoRE. Clearly, building a two-layer structure ("2-2" or "4-4") from a flat layer of experts ("4" or "8") boosts the accuracy without requiring additional trainable parameters. 4. Finally, the comparable parameter counts of MixLoRA and S'MoRE implies that our multi-layer design introduces low parameter overhead, which is consistent with our analysis in §3.4.

Table 3: LLaMA 3-8B: model Accuracy / Pass@1, and the best-performing models' trainable parameters (B).

| Gate | Method | GSM8K Accuracy | GSM8K Param. (B) | HumanEval Pass@1 | HumanEval Param. (B) |
|---|---|---|---|---|---|
| | Base model | 55.95 | 0 | 26.22 | 0 |
| | LoRA | 59.97 | 0.014 | 43.29 | 0.014 |
| Dense | HydraLoRA (4) | 62.47 | 0.317 | 40.85 | 0.082 |
| | HydraLoRA (8) | 62.24 | 0.297 | **44.51** | 0.079 |
| | MixLoRA (4) | 61.11 | 0.132 | 39.02 | 0.026 |
| | MixLoRA (8) | 59.36 | 0.132 | 40.85 | 0.033 |
| | S'MoRE (2-2) | 62.40 | 0.104 | 42.07 | 0.090 |
| | S'MoRE (4-4) | **65.20** | 0.957 | 43.90 | 0.104 |
| Switch | MixLoRA (4) | 59.67 | 0.047 | 42.68 | 0.075 |
| | MixLoRA (8) | 61.56 | 0.247 | 39.63 | 0.247 |
| | S'MoRE (2-2) | 62.47 | 0.133 | **45.73** | 0.190 |
| | S'MoRE (4-4) | **63.91** | 0.957 | 42.07 | 0.090 |

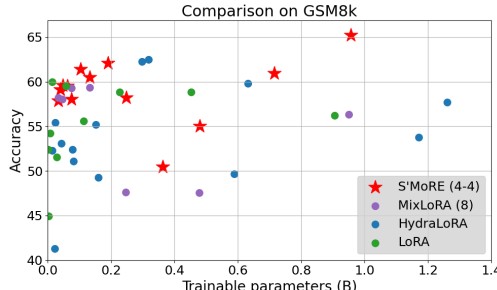

Figure 4: Change of accuracy w.r.t. trainable parameters, corresponding to models in Table 3.

Table 4: Results on Gemma 2-9B. We evaluate on representative benchmarks due to limited resources.

| Method | ARC-e | | CSQA | | Winogrande | | HumanEval | | Avg | Avg |
| | Accuracy | Param. (B) | Accuracy | Param. (B) | Accuracy | Param. (B) | Pass@1 | Param. (B) | Acc. / Pass@1 | Param. (B) |
|---|---|---|---|---|---|---|---|---|---|---|
| LoRA | 79.72 | 0.289 | 85.91 | 0.145 | 87.06 | 0.145 | 43.29 | 0.072 | 74.00 | 0.163 |
| MixLoRA (4) | 85.54 | 0.059 | 85.83 | 0.096 | 88.79 | 0.169 | 43.29 | 0.096 | 75.86 | 0.105 |
| MixLoRA (8) | 83.07 | 0.168 | 85.83 | 0.096 | 89.19 | 0.315 | 44.51 | 0.168 | 75.65 | 0.187 |
| S'MoRE (2-2) | 86.24 | 0.042 | **86.40** | 0.169 | **90.13** | 0.169 | 44.51 | 0.096 | 76.82 | 0.119 |
| S'MoRE (4-4) | **86.60** | 0.169 | 86.32 | 0.060 | **90.13** | 0.315 | **46.34** | 0.060 | **77.35** | 0.151 |

Table 5: S'MoRE on LLaMA 3.2-1B with more layers. We follow a simple hyperparameter tuning strategy, ensuring the same design space sizes and parameter budgets for the 2- and 3-layer variants.

| Layer sizes | ARC-c | | ARC-e | | Commonsense QA | | OpenBook QA | | Winogrande | |
| | Accuracy | Param. (B) | Accuracy | Param. (B) | Accuracy | Param. (B) | Accuracy | Param. (B) | Accuracy | Param. (B) |
|---|---|---|---|---|---|---|---|---|---|---|
| 2-2 | **40.00** | 0.017 | **75.31** | 0.085 | 66.99 | 0.037 | 72.20 | 0.085 | 52.01 | 0.011 |
| 2-2-2 | 39.32 | 0.017 | 74.25 | 0.102 | **67.40** | 0.053 | **72.60** | 0.205 | **52.88** | 0.011 |
| 4-4 | 39.66 | 0.017 | **74.43** | 0.085 | **67.32** | 0.045 | 72.80 | 0.202 | 52.01 | 0.168 |
| 4-4-4 | **40.34** | 0.029 | 73.90 | 0.205 | **67.32** | 0.053 | **73.60** | 0.202 | **52.09** | 0.013 |

### 4.3 Results on GSM8K & HumanEval

We evaluate on GSM8K and HumanEval using LLaMA 3-8B. The observations on accuracy / Pass@1 and parameter efficiency from Table 3 is consistent with those from Table 2: S'MoRE achieves significant accuracy improvement while maintaining parameter efficiency. Fig. 4 helps us better understand how the model accuracy scales with the amount of trainable parameters. 1) For S'MoRE, the accuracy consistently increases with parameters in the low-parameter region (less then 0.2B). Then the accuracy drops when we keep increasing the parameters. Interestingly, in the region from 0.4B to 1B, we see an almost linear increase of accuracy w.r.t. parameters – the accuracy eventually surpasses that of all other models with a large margin at around 1B. 2) For HydraLoRA, its accuracy peaks at around 0.3B. Unlike S'MoRE, keeping increasing the parameters does not help with HydraLoRA's accuracy improvement. This observation is consistent with Table 2. 3) Similar to HydraLoRA, the 1-layer MixLoRA does not show good scaling of accuracy w.r.t. parameters. S'MoRE may discover good structures among experts, which in turn helps experts better utilize their parameters.

### 4.4 Evaluation on Gemma

We extend our evaluation to the Gemma model family. Table 4 shows the comparison with representative baselines. Consistent with the observations on the LLaMA family, S'MoRE achieves significant boost in accuracy / Pass@1 with comparable or fewer parameters (see "MixLoRA (4) *vs.* S'MoRE (2-2)" and "MixLoRA (8) *vs.* S'MoRE (4-4)"). The performance gains across multiple model scales (1B, 7B, 9B) and model families (LLaMA, Gemma) reaffirm the benefits from structural mixture.

### 4.5 Scaling up with Layers

We evaluate if increasing the number of S'MoRE layers can further improve accuracy. We follow a simple hyperparameter tuning strategy: for all the 2-layer S'MoRE under consideration, we add a $3^{rd}$ layer with identical configuration (w.r.t. number of experts $s$, fanout $f$, expert dimension $r$, etc.) as the $2^{nd}$ layer. Thus, the sizes of the design spaces for the 3-layer and 2-layer S'MoRE are equal. We also enforce the same parameter budget for the 2- and 3-layer models. Table 5 summarizes the comparison. Adding one more layer improves accuracy significantly in many cases. The accuracy improvements do not necessarily come at the cost of more parameters. For example, for Winogrande, "2-2-2" structure improves the accuracy of "2-2" by 0.87 with the same parameter count.

## 5 Conclusion

We introduced S'MoRE, a novel Structural Mixture of Residual Experts framework that jointly achieves the efficiency of low-rank adaptation (LoRA) with the flexibility of Mixture-of-Experts (MoE), and further boosts MoE's model capacity by exploiting experts' inherent structure. By applying hierarchical residual decomposition and tree-based routing, S'MoRE effectively emulates exponentially more experts without instantiating additional expert instances, and achieves similar computation and parameter efficiency as the vanilla LoRA. We further propose a structural flexibility metric to quantify the model capacity, and theoretically show that S'MoRE's unique model architecture design is the key to boost structural flexibility compared with various LoRA-MoE hybrids. On extensive experiments, we confirm S'MoRE's state-of-the-art fine-tuning performance.

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

## A  More Related Works

**More related works on MoE.** In addition to the works mentioned in §2, we list a few additional works that explore MoE designs in LLM. LLaMA-MoE [Zhu et al., 2024] applies Mixture-of-Experts under the Continual Pre-training (CPT) setting. It breaks down LLaMA's pre-trained weight matrices into sub-matrices and use them to initialize the experts' parameters. MC-SMoE [Li et al., 2024b] discovers that there exists high redundancy among experts, and correspondingly proposes algorithms to cluster, merge and then compress a model with many experts. MoA [Feng et al., 2024] explores Mixture-of-LoRAs for the multi-task tuning scenarios, where it trains a LoRA for each task separately, and then assembles the multiple LoRAs into an MoE where a learnable router selects the most suitable expert based on the task category.

**Heterogeneous experts.** In most MoE models, experts are homogeneous and may have an identical model architecture. Heterogeneous experts have been recently explored in language modeling [Raposo et al., 2024, Ainslie et al., 2023] and graph learning [Zeng et al., 2024]. MoD [Raposo et al., 2024] and CoLT5 [Ainslie et al., 2023] consider the combination of light and heavy experts so that tokens through the light path can be processed faster and more cheaply. Mowst [Zeng et al., 2024] further discovers that mixture of weak and strong experts can enhance MoE's capacity beyond that of a strong expert alone. However, existing heterogeneous MoE designs consider a *horizontal* stacking of different types of experts, where the weak/light branch operates in parallel to the strong/heavy one. However, in S'MoRE, the residuals of different orders can also be interpreted as experts of different strength, with the 1st-order residuals being the strongest. And we explore a *vertical* stacking design where the higher-order residuals transform and propagate to the lower-order ones. The heterogeneous expert design in S'MoRE encodes additional structural information than existing models.

**Scaling behavior.** There is an emerging trend to research the unique scaling behaviors of MoE systems compared with dense models. Ludziejewski et al. [2024] summarizes a scaling law indicating that finer expert granularity may improve model capacity, and He [2024] provides a practical implementation for very large number of experts. Zhao et al. [2025] applies such fine-grained experts in the fine-tuning tasks. Through analysis and experiments, we hypothesize that "structural flexibility" may be a neglected yet critical factor impacting MoE scaling. S'MoRE has the potential to scale better than regular MoEs, for fine-tuning tasks and beyond. We leave the application of S'MoRE on other types of tasks as future work.

## B  Details of Model Design

### B.1  Three types of gates implemented in practice

In our experiments, we implement S'MoRE and the baselines with various types of dense and sparse gates, which we describe in more details here:

- *Dense* gate: Here we activate all children experts, meaning that $f_\ell = s_\ell$. For each children, the expert score $\alpha$ is still generated by the gating neural net. So the dense gate can be understood as a soft version of the sparse gates below. Now since all experts are activated, we do not include additional auxiliary loss for load-balance of all the experts. The same dense gate design is also used by HydraLoRA [Tian et al., 2024] which we include as one of the experimental baselines (see §4).

- *Sparse noisy top-k* gate [Shazeer et al., 2017]: For each layer $\ell$, the router selects the top $k = f_\ell$ children residuals with the highest gating scores (generated by Eq. 6). However, a common issue with sparse expert selection is expert under-utilization, where certain experts are overused while others remain idle, resulting in inefficient training dynamics. To address this, following [Shazeer et al., 2017], we first add a learnable noise term on top of gating score $\alpha$, to encourage exploration of different expert choices. Then we implement a layer-wise importance loss plus balance loss, computed separately for each set of experts $\mathcal{R}_\ell$ based on gating score distribution and activation frequency. We sum up the two auxiliary losses of each S'MoRE layer and each transformer layer, and add it to the model prediction cross-entropy (CE) loss, i.e.,

$$\mathcal{L}_{\text{total}} = \mathcal{L}_{\text{CE}} + \gamma \cdot \sum_{\ell_{\text{trans}}=1}^{L_{\text{trans}}} \sum_{\ell=1}^{L} \left( \mathcal{L}_{\ell_{\text{trans}},\ell}^{\text{importance}} + \mathcal{L}_{\ell_{\text{trans}},\ell}^{\text{balance}} \right), \tag{11}$$

where $L_{\text{trans}}$ denotes the total number of transformer layers and $\gamma$ is a coefficient ($\ll 1$) controlling the strength of the load-balance constraint. The importance and balance loss computation exactly follows the original paper (see Section 4 and Appendix A of Shazeer et al. [2017]).

- *Sparse switch-transformer* gate [Fedus et al., 2022b]: This is another popular sparse gate design. There are two main differences from the noisy top-$k$ gate above. First, the switch gate implements an optional jitter noise that is applied to the gate's input embedding rather than the final gating score. In addition, the switch gate integrates a different way to compute the load-balance auxiliary loss. The auxiliary losses of all transformer and S'MoRE layers are still added back to the CE loss of the main task, just like Eq. 11. To ensure fair comparison in experiments, we exactly follow the balance loss implementation in the MixLoRA codebase[3].

### B.2  Alternative design: bottom-up routing

We propose the following bottom-up router fully compatible with S'MoRE's structural mixing design.

In bottom-up routing, our goal remains the same as the top-down router of §3.3: to customize different children experts for different parents. Yet, under bottom-up routing, the parent index is unknown when we decide the children. So in our design, we will still include a key vector $\boldsymbol{k}$ (see Eq. 6). Yet this $\boldsymbol{k}$ is not directly associated with any specific parent expert. It represents a node position in the routing tree.

To avoid the discussion being overwhelmed by notations, we use the following example to illustrate the core idea. Consider a 3-layer S'MoRE, where each layer has a fanout of $f = 2$, and has $s = 6$ experts. Like our original design, the router still has the following:

---

[3]`https://github.com/TUDB-Labs/MixLoRA`

- a down-projection matrix that projects the original token $x$ to $d_{\text{down}}$-dimensional $x_{\text{down}}$ (e.g., $d_{\text{down}} = 16$);

- a small MLP in each layer (except layer 1).

In addition, we instantiate a learnable "key" tensor $K$ in each layer. $K$'s last dimension is $d_{\text{down}} = 16$, and $K$'s leading dimensions depend on the fanout in parent layers (except that for layer 1, $K$'s dimension additionally depends on the number of experts). For example, in layer 1, $K$ has shape $(2, 2, 6, 16)$. In layer 2 and 3, $K$'s shapes are $(2, 2, 16)$ and $(2, 16)$.

The routing proceeds as follows:

1. At layer 1, we compute dot product between $K$ and $x_{\text{down}}$ along the last dimension, generating a score tensor of shape $(2, 2, 6)$. Taking the top-2 along the last dimension, we determine the $2 \times 2 \times 2$ children experts for all the $2 \times 2$ parents.

2. Then we follow Eq. 3 to aggregate the children experts' outputs, generating $2 \times 2$ different output embeddings.

3. In layer 2, we concatenate the $2 \times 2$ layer-1 output embeddings and the $(2, 2, 16)$-shaped $K$, along the last dimension. Then we feed the concatenated tensor into layer 2's router MLP to generate the score distribution over all 6 candidate experts. Taking the top-2 along the last dimension of the MLP output, we now select the $2 \times 2$ layer-2 experts, conditioned on the layer-1 children.

4. Layer 3 operates similarly.

**Comparison.** From the above, it is clear that the bottom-up router is still computationally efficient (similar to the original top-down router). Both the bottom-up and top-down designs can model the interaction between parent and children experts. The main difference is that the bottom-up router makes routing decisions based on the children's aggregated embedding, while the top-down router directly consumes the token embedding. So in the bottom-up design, the gradient can flow back to the lower-layer experts from the router. This may lead to interesting training behaviors that differ from the top-down case.

Due to limited GPUs resources, we are unable to run ablation with the bottom-up router. We leave such evaluation as an important future work.

## C  Details of Theoretical Analysis

### C.1  Derivation of parameter & computation costs

Here we provide additional algorithmic details for the parameter and computation efficiency calculation in §3.4.

**Parameter efficiency.** From Eq. 8, we have

$$P_{\ell+1} = s_\ell \cdot d \cdot r_\ell + d_{\ell+1}^2 \tag{12}$$

Then Eq. 9 is derived as

$$P_{\text{proj}} + \sum_{\ell=1}^{L} P_\ell = d \cdot d_L + \sum_{\ell=0}^{L-1} P_{\ell+1}$$

$$= d \cdot d_L + \sum_{\ell=0}^{L-1} s_\ell \cdot d \cdot r_\ell + \sum_{\ell=0}^{L-1} d_{\ell+1}^2$$

$$= d \cdot d_L + d \cdot \left( \sum_{\ell=0}^{L-1} s_\ell \cdot r_\ell \right) + \sum_{\ell=0}^{L-1} d_{\ell+1}^2$$

$$= d \cdot d_L + d \cdot d_L + \sum_{\ell=0}^{L-1} d_{\ell+1}^2$$

$$= 2 \cdot d \cdot d_L + \Delta \tag{13}$$

$$\text{where} \qquad \Delta = \sum_{\ell=0}^{L-1} d_{\ell+1}^2 \ll 2 \cdot d \cdot d_L$$

**Computation cost.** In Eq. 3, each $\boldsymbol{B}_\ell^n \cdot \boldsymbol{A}_\ell^n \cdot \boldsymbol{x}$ requires $C' = d \cdot r_\ell + r_\ell \cdot d_{\ell+1}$ operations. Each $\boldsymbol{W}_\ell \cdot \boldsymbol{x}_\ell^n$ requires $C'' = d_\ell \cdot d_{\ell+1}$. Consider all activated experts $i$ in layer $\ell + 1$, there can be at most $N' = \min\{s_\ell, F_\ell\}$ distinct $\boldsymbol{B}_\ell^n \cdot \boldsymbol{A}_\ell^n$ terms, incurring $C'$ cost $N'$ times. There are $F_\ell$ different $\boldsymbol{x}_\ell^n$ inputs, each incurring $C''$ cost. Ignoring the element-wise addition "$\sum_{n \in \mathcal{N}_\ell^i}$" and multiplication $\alpha_\ell^{i,n}$, the total cost of layer $\ell + 1$ equals (where $d_\ell$ and $F_\ell$ follow Eq. 4 and Eq. 5):

$$C_{\ell+1} \leq \min\{s_\ell, F_\ell\} \cdot r_\ell \cdot (d + d_{\ell+1}) + F_\ell \cdot d_\ell \cdot d_{\ell+1}$$

$$= \min\{s_\ell, F_\ell\} \cdot r_\ell \cdot d + \min\{s_\ell, F_\ell\} \cdot r_\ell \cdot d_{\ell+1} + F_\ell \cdot d_\ell \cdot d_{\ell+1}$$

$$\leq s_\ell \cdot r_\ell \cdot d + F_\ell \cdot r_\ell \cdot d_{\ell+1} + F_\ell \cdot d_\ell \cdot d_{\ell+1}$$

$$= s_\ell \cdot r_\ell \cdot d + F_\ell \cdot d_{\ell+1} \cdot (d_\ell + r_\ell) \tag{14}$$

The cost of the final projection equals $C_{\text{proj}} = d \cdot d_L$. So the overall computation cost is:

$$C_{\text{proj}} + \sum_{\ell=1}^{L} C_\ell = d \cdot d_L + d \cdot \left( \sum_{\ell=0}^{L-1} s_\ell \cdot r_\ell \right) + \Delta' \overset{(b)}{=} 2 \cdot d \cdot d_L + \Delta' \overset{(c)}{\approx} 2 \cdot d \cdot d_L \tag{15}$$

where $\Delta' \leq \sum_{\ell=0}^{L-1} F_\ell \cdot d_{\ell+1} \cdot (d_\ell + r_\ell)$. Steps "(b)" and "(c)" follow similar reasoning to Eq. 9.

Under practical values of $d_\ell + r_\ell \leq d_{\ell+1} \ll d$, the overhead term $\Delta'$ is small or negligible compared to the main cost $2 \cdot d \cdot d_L$. In Table 6, we empirically calculate the value of $2 \cdot d \cdot d_L$, the overhead $\Delta'$, and their ratio. We take a representative configuration with $f_\ell = 2$, $s_\ell = 4$, and $r_\ell = 8$ or $16$ for all layers $\ell$ (consistent with the experiments in §4). For 2 layers, the overhead $\Delta'$ is just **1.2%** (or **2.3%**) of the cost of vanilla LoRA with rank $d_L = 64$ (or $d_L = 128$).

Table 6: Overhead $\Delta'$ compared with the main computation cost $2 \cdot d \cdot d_L$

Similar to the analysis in the "Parameter efficiency" paragraph, the gating MLPs are lightweight compared to the main cost $2 \cdot d \cdot d_L$, due to the small dimensionalities.

Thus, the total computation cost of S'MoRE is approximately $2 \cdot d \cdot d_L$, which is *the same as the cost of a vanilla LoRA with rank $d_L$*.

| $r_\ell$ | $L$ | $d_L$ | $2 \cdot d \cdot d_L$ | $\Delta'$ | Overhead ratio |
|---|---|---|---|---|---|
| | 2 | 64 | 0.5M | 0.006M | 1.2% |
| 8 | 3 | 96 | 0.8M | 0.026M | 3.3% |
| | 4 | 128 | 1.0M | 0.079M | 7.5% |
| | 2 | 128 | 1.0M | 0.025M | 2.3% |
| 16 | 3 | 192 | 1.6M | 0.104M | 6.6% |
| | 4 | 256 | 2.1M | 0.315M | 15.0% |

with rank $d_L$. This proves the both parameter and the computation efficiency of S'MoRE.

## C.2 Proof of model capacity

### C.2.1 Proof for two special S'MoRE configurations

**Proposition C.1.** *(Proposition 3.1) S'MoRE can express MoLRE, when $L = 1$ and $\sigma(\cdot)$ is the identity mapping.*

*Proof.* When $L = 1$, there is only a single layer propagation. When we set $\sigma$ as the identity mapping, Eq. 3 becomes

$$x_1 = \sum_{n \in \mathcal{N}_0} \alpha_\ell^n \cdot \boldsymbol{B}_0^n \cdot \boldsymbol{A}_0^n \cdot \boldsymbol{x} \tag{16}$$

where we omit the superscript $i$ since there is just one parent node (the root of all all experts in a flat layer).

Combined with the final projection (see end of §3.2), the final output is computed by

$$\begin{aligned}
\boldsymbol{x}' &= \boldsymbol{W}_{\text{proj}} \cdot \sum_{n \in \mathcal{N}_0} \alpha_\ell^n \cdot \boldsymbol{B}_0^n \cdot \boldsymbol{A}_0^n \cdot \boldsymbol{x} \\
&= \sum_{n \in \mathcal{N}_0} \alpha_\ell^n \cdot (\boldsymbol{W}_{\text{proj}} \cdot \boldsymbol{B}_0^n) \cdot \boldsymbol{A}_0^n \cdot \boldsymbol{x}
\end{aligned} \tag{17}$$

where $\boldsymbol{A}_0^n \in \mathbb{R}^{r_0 \times d}$, $\boldsymbol{B}_0^n \in \mathbb{R}^{(s_0 \cdot r_0) \times r_0}$ and $\boldsymbol{W}_{\text{proj}} \in \mathbb{R}^{d \times (s_0 \cdot r_0)}$.

For MoLRE with $s_0$ rank-$r_0$ experts, according to the definition in §3.1, we can express its layer operation as

$$\bar{\boldsymbol{x}}' = \sum_{n \in \mathcal{N}} \texttt{ROUTE}\,(\boldsymbol{x})^n \cdot \bar{\boldsymbol{B}}^n \cdot \bar{\boldsymbol{A}}^n \cdot \boldsymbol{x} \tag{18}$$

where we use overhead "bar" to distinguish variables of MoLRE from those of 1-layer S'MoRE. Here $\bar{\boldsymbol{A}}^n \in \mathbb{R}^{r_0 \times d}$ and $\bar{\boldsymbol{B}}^n \in \mathbb{R}^{d \times r_0}$.

To make Eq. 17 and Eq. 18 equivalent, we can have

- S'MoRE's router implementing as $\texttt{ROUTE}\,(\boldsymbol{x})^n$

- $\boldsymbol{A}_0^n = \bar{\boldsymbol{A}}^n$ (by definition, both matrices have the same shape)

- $\boldsymbol{B}_0^n = \begin{bmatrix} \boldsymbol{0}_{r_0} \\ \vdots \\ \boldsymbol{0}_{r_0} \\ \boldsymbol{I}_{r_0} \\ \boldsymbol{0}_{r_0} \\ \vdots \\ \boldsymbol{0}_{r_0} \end{bmatrix}$, which is a binary matrix by vertically stacking $s_0$ square blocks of $r_0 \times r_0$

  sub-matrices. The $n$-th block is a $r_0 \times r_0$ identity matrix, $\boldsymbol{I}_{r_0}$, while all the other blocks are 0 (denoted as $\boldsymbol{0}_{r_0}$).

- $\boldsymbol{W}_{\text{proj}} = \left[\bar{\boldsymbol{B}}^1, \ldots, \bar{\boldsymbol{B}}^{s_0}\right]$.

Then $\boldsymbol{W}_{\text{proj}} \cdot \boldsymbol{B}_0^n = \bar{\boldsymbol{B}}_0^n$. And Eq. 17 becomes identical to Eq. 18, completing the proof.

$\square$

**Proposition C.2.** *(Proposition 3.2) S'MoRE can express MoMOR, when setting $\sigma\,(\cdot)$ as the identity mapping.*

*Proof.* Without $\sigma$, we can collapse a multi-layer S'MoRE into a single-layer equivalent. For $L = 2$, following Eq. 3, we have

$$\boldsymbol{x}_2 = \sum_{n \in \mathcal{N}_1} \alpha_1^n \cdot (\boldsymbol{B}_1^n \cdot \boldsymbol{A}_1^n \cdot \boldsymbol{x} + \boldsymbol{W}_1 \cdot \boldsymbol{x}_1^n)$$

$$= \sum_{n \in \mathcal{N}_1} \alpha_1^n \cdot \boldsymbol{B}_1^n \cdot \boldsymbol{A}_1^n \cdot \boldsymbol{x} + \boldsymbol{W}_1 \sum_{n \in \mathcal{N}_1} \alpha_1^n \cdot \left( \sum_{m \in \mathcal{N}_0^n} \alpha_0^{n,m} \cdot \boldsymbol{B}_0^m \cdot \boldsymbol{A}_0^m \cdot \boldsymbol{x} \right)$$

$$= \sum_{n \in \mathcal{N}_1} \hat{\alpha}_1^n \cdot \boldsymbol{B}_1^n \cdot \boldsymbol{A}_1^n \cdot \boldsymbol{x} + \sum_{m \in \mathcal{N}_0} \hat{\alpha}_0^m \cdot (\boldsymbol{W}_1 \cdot \boldsymbol{B}_0^m \cdot \boldsymbol{A}_0^m) \cdot \boldsymbol{x} \qquad (19)$$

where we define $\hat{\alpha}_1^n = \alpha_1^n$ and $\hat{\alpha}_0^m = \sum_{n \in \mathcal{N}_1 \text{ and } m \in \mathcal{N}_0^n} (\alpha_1^n \cdot \alpha_0^{n,m})$.

In general, for $L$ layers and with the final projection step $\boldsymbol{W}_{\text{proj}}$, we can summarize the propagation equation as

$$\boldsymbol{x}' = \sum_{\ell=0}^{L-1} \sum_{i=1}^{s_\ell} \hat{\alpha}_\ell^i \cdot \left( \prod_{k=\ell+1}^{L} \boldsymbol{W}_k \right) \cdot \boldsymbol{B}_\ell^i \cdot \boldsymbol{A}_\ell^i \cdot \boldsymbol{x} \qquad (20)$$

where we define $\boldsymbol{W}_L = \boldsymbol{W}_{\text{proj}} \in \mathbb{R}^{d \times d_L}$ and $\hat{\alpha}_\ell^i$ is a scalar coefficient by aggregating the router weights along all paths that end at the layer-$(\ell+1)$ expert $i$[4]. In other words, $\hat{\alpha}_\ell^i$ generalizes the definition of $\hat{\alpha}_0^m$ above. The "path" here refers to the "ancestral path" (Definition C.5) ending at $i$. See more discussion on the routing tree in Appendix C.2.3. Also, if an expert is never selected, we let its $\hat{\alpha}_\ell^i = 0$. This way, we can replace the summation over $\mathcal{N}_\ell$ in Eq. 19 with the summation over $1 \le i \le s_\ell$ in Eq. 20.

For MoMOR model, following Eq. 2, we write its layer propagation as

$$\boldsymbol{x}' = \sum_{\ell=1}^{L-1} \sum_{i=1}^{s_\ell} \texttt{ROUTE}_\ell (\boldsymbol{x})^i \cdot \bar{\boldsymbol{B}}_\ell^i \cdot \bar{\boldsymbol{A}}_\ell^i \cdot \boldsymbol{x} \qquad (21)$$

We can make Eq. 20 and Eq. 21 equivalent by a similar construction as the proof for Proposition 3.1. First, define a special binary projection matrix $\boldsymbol{P}_{a \times b} \in \{0,1\}^{a \times b}$ (where $a > b$) as

$$\boldsymbol{P}_{a \times b} = \begin{bmatrix} \boldsymbol{0}_{(a-b) \times b} \\ \boldsymbol{I}_{b \times b} \end{bmatrix} \qquad (22)$$

meaning that the first $a - b$ rows of $\boldsymbol{P}_{a \times b}$ are all 0, and the bottom $b$ rows are an identity matrix. It is easy to verify that for $a > b > c$:

$$\boldsymbol{P}_{a \times b} \cdot \boldsymbol{P}_{b \times c} = \boldsymbol{P}_{a \times c} \qquad (23)$$

Then we can set all parameters of S'MoRE as follows:

- Let the S'MoRE router implement $\texttt{ROUTE}_\ell (\boldsymbol{x})^i$.

- Let $\boldsymbol{A}_\ell^i = \bar{\boldsymbol{A}}_\ell^i$.

- Let $\boldsymbol{B}_\ell^i$ be a $d_{\ell+1} \times r_\ell$ binary matrix, where its row $(i-1) \cdot r_\ell + 1$ to row $i \cdot r_\ell$ is a $r_\ell \times r_\ell$ identity matrix, and its all other rows are all 0. Here we let both $i$ and the row index start from 1.

- Let $\boldsymbol{W}_L = \boldsymbol{W}_{\text{proj}} = \left[ \bar{\boldsymbol{B}}_0^1, \ldots, \bar{\boldsymbol{B}}_0^{s_0}, \ldots, \bar{\boldsymbol{B}}_{L-1}^1, \ldots, \bar{\boldsymbol{B}}_{L-1}^{s_{L-1}} \right]$ as the horizontal concatenation of all MoMOR's up-projection matrices $\bar{\boldsymbol{B}}_\ell^i$.

---

[4]The same expert $i$ of layer $\ell+1$ may be selected multiple times, corresponding to different parents or ancestors. Thus, there can be multiple paths ending at the layer-$(\ell+1)$ expert $i$.

- Each $\boldsymbol{W}_k$ has shape $d_{k+1} \times d_k$ where $d_{k+1} = d_k + s_k \cdot r_k$. We set it as $\boldsymbol{W}_k = \boldsymbol{P}_{d_{k+1} \times d_k}$. Then it follows that

$$\prod_{k=\ell+1}^{L-1} \boldsymbol{W}_k = \boldsymbol{P}_{d_L \times d_{L-1}} \cdot \boldsymbol{P}_{d_{L-1} \times d_{L-2}} \ldots \boldsymbol{P}_{d_{\ell+2} \times d_{\ell+1}} = \boldsymbol{P}_{d_L \times d_{\ell+1}}$$
(24)

$$\Rightarrow \quad \left( \prod_{k=\ell+1}^{L} \boldsymbol{W}_k \right) \cdot \boldsymbol{B}_\ell^i = \boldsymbol{W}_{\mathrm{proj}} \cdot \left( \prod_{k=\ell+1}^{L-1} \boldsymbol{W}_k \right) \cdot \boldsymbol{B}_\ell^i$$
$$= \boldsymbol{W}_{\mathrm{proj}} \cdot \boldsymbol{P}_{d_L \times d_{\ell+1}} \cdot \boldsymbol{B}_\ell^i$$
$$= \bar{\boldsymbol{B}}_\ell^i$$
(25)

Under the above construction, it is clear that Eq. 21 and Eq. 20 are exactly the same. Thus, S'MoRE can express MoMOR, concluding the proof.

**Remark.** Note that the equivalence between S'MoRE and MoMOR can only be established when we set the layer dimension $d_\ell$ according to Eq. 4. This can be seen from the "minimum dimensionality" discussion in §3.2.

$\square$

### C.2.2 Proof of Theorem 3.3

**Theorem C.3.** *(Theorem 3.3) The structural flexibility of MoMOR is upper-bounded by* $\Gamma_{MoMOR} = \max_{\boldsymbol{x},\Theta} \mathit{dist}\,(\boldsymbol{x};\Theta) \leq \binom{s_{L-1}}{f_{L-1}} \cdot \prod_{\ell=0}^{L-2} \left( \sum_{i=f_\ell}^{\min\{F_\ell, s_\ell\}} \binom{s_\ell}{i} \right).$

*Proof.* The upper bound of $\Gamma_{\mathrm{MoMOR}}$ basically quantifies the total number of combinations to select experts from each residual pool.

**Assumption.** We first simplify Eq. 2 that the router-generated coefficient $\mathtt{ROUTE}_\ell\,(\boldsymbol{x})^i$ is just a binary mask. i.e., for a selected expert $i$, we have $\mathtt{ROUTE}_\ell\,(\boldsymbol{x})^i = 1$. Otherwise, $\mathtt{ROUTE}_\ell\,(\boldsymbol{x})^i = 0$. Such an assumption is just to ease the calculation of $\Gamma_{\mathrm{MoMOR}}$ and $\Gamma_{\mathrm{S'MoRE}}$. It does not affect our fundamental conclusion that S'MoRE yields exponentially higher structural flexibility than MoMOR.

Based on Eq. 2, the MoMOR output is generated by a flat summation of different-order residues. Given any input $\boldsymbol{x}$, the number of distinct outputs cannot exceed the number of distinct ways to select residues from the pools $\mathcal{R}_0, \cdots, \mathcal{R}_{L-1}$. Here we show some examples to illustrate the meaning of "distinct expert selection".

- "Selecting experts 1,2,3 from $\mathcal{R}_0$" and "selecting experts 1,3,4 from $\mathcal{R}_0$" correspond to 2 distinct ways.

- "Selecting experts 1,2,3 from $\mathcal{R}_0$" and "selecting experts 3,2,1 from $\mathcal{R}_0$" correspond to the same way, because there is no ordering among the selected experts[5].

- "Selecting experts 1,1,3 from $\mathcal{R}_0$" and "selecting experts 1,3,3 from $\mathcal{R}_0$"[6] correspond to the same way due to our assumption of making $\mathtt{ROUTE}_\ell\,(\boldsymbol{x})^i$ a binary mask. Basically we only care about whether an expert is selected or not. It does not matter how many times an expert is selected.

**Remark.** Distinct expert selections do not guarantee distinct outputs. For example, consider "selecting 1,2,3 from $\mathcal{R}_0$" and "selecting 1,3,4" from $\mathcal{R}_0$. Following the notation of Eq. 2, if the experts' weights satisfy $\Delta \boldsymbol{W}_0^1 + \Delta \boldsymbol{W}_0^2 + \Delta \boldsymbol{W}_0^3 = \Delta \boldsymbol{W}_0^1 + \Delta \boldsymbol{W}_0^3 + \Delta \boldsymbol{W}_0^4$, then the two case generates the same output for all input $\boldsymbol{x}$:

---

[5]The order among selected experts does not matter because the sum aggregation of Eq. 2 is *permutation invariant*

[6]If we follow S'MoRE's recursive expert selection process described in §3.3, the same expert of higher-order may be selected multiple times, from different lower-order parents.

$$\sum_{i \in \{1,2,3\}} \Delta \boldsymbol{W}_0^i \cdot \boldsymbol{x} = \sum_{j \in \{1,3,4\}} \Delta \boldsymbol{W}_0^j \cdot \boldsymbol{x} \tag{26}$$

Hence, counting the number of distinct ways of expert selection just gives an upper bound of $\Gamma_{\text{MoMOR}}$, because $\Gamma_{\text{MoMOR}}$ is defined on the number of distinct outputs.

**Counting the combinations.** For the $\mathcal{R}_{L-1}$ pool with size $s_{L-1}$, there are $\binom{s_{L-1}}{f_{L-1}}$ ways to pick $f_{L-1}$ residues. For $\mathcal{R}_{L-2}$ with $\ell \leq L - 2$, there are $F_{\ell+1}$ parents, each picking $f_\ell$ children in the pool. Different parents can pick the same children. The number of distinct children selected by all parents ranges from $f_\ell$ to $\min\{F_\ell, s_\ell\}$. This makes the total count $\sum_{i=f_\ell}^{\min\{F_\ell, s_\ell\}} \binom{s_\ell}{i}$. From basic Combinatorics, each layer $\ell$ contributes to a multiplicative factor in the total count. Thus, the final upper bound is:

$$\Gamma_{\text{MoMOR}} \leq \binom{s_{L-1}}{f_{L-1}} \cdot \prod_{\ell=0}^{L-2} \left( \sum_{i=f_\ell}^{\min\{F_\ell, s_\ell\}} \binom{s_\ell}{i} \right) \tag{27}$$

$\square$

### C.2.3 Proof of Theorem 3.4

**Theorem C.4.** *(Theorem 3.4) Setting $\sigma(\cdot)$ as an MLP, there exists some $\Theta'$ such that the structural flexibility of* S'MoRE *is $\Gamma_{S'MoRE} = \min_{\boldsymbol{x}} dist(\boldsymbol{x}; \Theta') = \prod_{\ell=0}^{L-1} \binom{s_\ell}{f_\ell}^{F_{\ell+1}}$, where $F_L := 1$.*

*Proof.* We prove in two stages:

1. We show that following the routing process of S'MoRE, there can be $\Gamma_{S'MoRE}$ non-isomorphic depth-$L$ trees, where each tree node is an expert residue.

2. We construct a S'MoRE instance where its $L$-layer propagation (Eq. 3) generates distinct outputs for all non-isomorphic trees above, regardless of input token embedding $\boldsymbol{x}$.

Both can be proven by induction.

**Assumption.** Similar to Theorem 3.3, we make simplification to the layer propagation Eq. 3, that the coefficient $\alpha_\ell^{i,n}$ is just a binary mask. i.e., for a selected children $n$, we have $\alpha_\ell^{i,n} = 1$. Otherwise, $\alpha_\ell^{i,n} = 0$.

**Stage 1: Number of non-isomorphic trees.** Recall the expert selection / tree construction process in §3.3: each active parent expert of layer $\ell + 1$ (in $\mathcal{R}_\ell$) selects $f_{\ell-1}$ children out of all the $s_{\ell-1}$ experts of layer $\ell$. So by traversing all the $L$ layers, the router builds a depth-$L$ balanced tree (which has $\prod_{\ell=0}^{L-1} f_\ell$ leaf nodes in total). Note that

1. For each parent, its $f_\ell$ selected children are distinct (i.e., the same parent cannot select the same child twice).

2. However, the same expert may appear in the same tree-level multiple times, corresponding to different parents or ancestors.

3. There is **no ordering** among the selected children, since Eq. 3 performs "sum" aggregation which is *permutation invariant*. e.g., it is equivalent to say that a parent of layer $\ell$ selects "children 1,3,4" and "children 4,3,1".

Due to Point 2 above, we cannot uniquely identify a tree node by the its corresponding expert's layer index and expert index. Yet, Points 1 and 3 ensure that any tree node $n$ is *uniquely identifiable* by $n$'s ancestral path $\mathcal{P}_n$.

**Definition C.5.** (Ancestral path $\mathcal{P}_n$) Let $(\ell, i)$ denote expert $i$ of layer $\ell$. Suppose a tree-node $n$ at tree-level $t$ corresponds to expert $(L - t + 1, i)$. Then $n$'s ancestral path, $\mathcal{P}_n = ((L - t + 1, i), (L - t + 2, i'), \ldots, (L, i'^{\cdots'}))$, defines the unique path to traverse from $n$ up to the tree root (where we treat the root as a *virtual* node that is the parent of all $(L, i'^{\cdots'})$, and we omit the root in the path).

**Definition C.6.** (Leaves' ancestral paths $\mathcal{T}$) Given a tree, define $\mathcal{T} = \{\mathcal{P}_n \mid n \text{ is a leaf node}\}$ as the set of ancestral paths of all leaf nodes, where there are $\prod_{\ell=0}^{L-1} f_\ell$ leaves, all at tree-level $L$.

Two trees are *isomorphic* if their structures are equivalent. That means, we can permute or swap the children (together with their corresponding descendant sub-tree) of some parent nodes to make the two trees look exactly the same. $\mathcal{T}$ enables us to define isomorphism. In our construction, the children are not ordered (Point 3 above), and so permuting or swapping children does not change $\mathcal{T}$. Thus, isomorphic trees have the same $\mathcal{T}$. On the other hand, we can show trees of the same $\mathcal{T}$ can be made equivalent by permutation or swapping, and thus are isomorphic. In sum, we can define tree isomorphism by $\mathcal{T}$ as follows:

**Definition C.7.** (Isomorphism) Given two trees, let their leaves' ancestral paths be $\mathcal{T}$ and $\mathcal{T}'$. The two trees are isomorphic if and only if $\mathcal{T} = \mathcal{T}'$.

We next derive the number of depth-$L$ non-isomorphic trees by induction.

Imagine that we apply the top-down expert selection from layer $\ell$ down to layer 1 (with $\ell \geq 1$): at layer $\ell$, we select $f_{\ell-1}$ experts from $s_{\ell-1}$ experts; at layer $\ell - 1$, for each of the selected parent of layer $\ell$, we select $f_{\ell-2}$ from $s_{\ell-2}$ experts, and so on.

*Induction hypothesis*: the number of non-isomorphic trees yielded by such an expert-selection process equals:

$$\Gamma_{\text{S,MoRE}}^\ell = \prod_{k=0}^{\ell-1} \binom{s_k}{f_k}^{F_{k+1}/F_\ell} \tag{28}$$

for some $1 \leq \ell < L$.

*Base case $\ell = 1$*: we are just sampling a single level. So the number of non-isomorphic trees equals the number of total ways to select $f_0$ experts from $s_0$, which is $\binom{s_0}{f_0}$.

And

$$\Gamma_{\text{S,MoRE}}^1 = \prod_{k=0}^{1-1} \binom{s_k}{f_k}^{F_{k+1}/F_1}$$
$$= \binom{s_0}{f_0} \tag{29}$$

So the base case holds.

*Induction from $\ell$ to $\ell + 1$*: To construct a tree by selecting experts from layer $\ell + 1$ to 1, we follow two steps:

1. We select $f_\ell$ out of $s_\ell$ experts. Denote them as $\mathcal{E}_\ell = \{(\ell + 1, i_1), \ldots, (\ell + 1, i_{f_\ell})\}$, where $i_a \neq i_b$ for all $a \neq b$.

2. We start from each $(\ell + 1, i_m)$ and recursively activate experts from layer $\ell$ down to 1 (where $1 \leq m \leq f_\ell$), following the procedure described above. Denote each such tree by its leaves' ancestral paths, $\mathcal{T}_{\ell, i_m}$.

Let $\mathbb{T}_\ell$ be the set of all possible $\mathcal{T}_{\ell, i_m}$ — note that $\mathbb{T}_\ell$ does not have subscript $i_m$, since an ancestral path ends at a virtual root node independent of $i_m$ (see Definition C.5), and thus $\mathbb{T}_\ell$ is the same for all $i_m$. Based on the induction hypothesis, $|\mathbb{T}_\ell| = \Gamma_{\text{S,MoRE}}^\ell$.

For such a tree constructed by the two steps above, let $\mathcal{T}_{\ell+1}$ be its leaves' ancestral paths:

$$\mathcal{T}_{\ell+1} = \bigcup_{k=1}^{f_\ell} \{p \oplus (\ell+1, i_k) \mid p \in \mathcal{T}_{\ell,i_k}\} \tag{30}$$

where "$\oplus$" means appending $(\ell+1, i_k)$ to the end of the path $p$. By Definition C.7, the total number of non-isomorphic trees equals the number of distinct $\mathcal{T}_{\ell+1}$, which can be calculated with the following reasoning:

- There are $\binom{s_\ell}{f_\ell}$ distinct ways to choose $\mathcal{E}_\ell$ of Step 1.

- For each choice of $\mathcal{E}_\ell$, there are $|\mathbb{T}_\ell|$ choices of $\mathcal{T}_{\ell,i_k}$ for each $i_k$ of $\mathcal{E}_\ell$, leading to $|\mathbb{T}_\ell|^{f_\ell}$ distinct combinations.

So the number of distinct $\mathcal{T}_{\ell+1}$ equals:

$$
\begin{aligned}
|\mathbb{T}_{\ell+1}| &= \binom{s_\ell}{f_\ell} \cdot |\mathbb{T}_\ell|^{f_\ell} \\
&= \binom{s_\ell}{f_\ell} \cdot \left(\Gamma_{\text{S,MoRE}}^{\ell}\right)^{f_\ell} \\
&= \binom{s_\ell}{f_\ell} \cdot \left(\prod_{k=0}^{\ell-1} \binom{s_k}{f_k}^{F_{k+1}/F_\ell}\right)^{f_\ell} \\
&= \binom{s_\ell}{f_\ell} \cdot \prod_{k=0}^{\ell-1} \binom{s_k}{f_k}^{F_{k+1} \cdot \frac{f_\ell}{F_\ell}} \\
&= \binom{s_\ell}{f_\ell}^{F_{\ell+1}/F_{\ell+1}} \cdot \prod_{k=0}^{\ell-1} \binom{s_k}{f_k}^{F_{k+1}/F_{\ell+1}} \\
&= \prod_{k=0}^{\ell} \binom{s_k}{f_k}^{F_{k+1}/F_{\ell+1}} \\
&= \Gamma_{\text{S,MoRE}}^{\ell+1} \tag{31}
\end{aligned}
$$

This completes the induction step. Thus, the total number of non-isomorphic trees for all $L$ layers equals $\Gamma_{\text{S,MoRE}}^{L} = \prod_{\ell=0}^{L-1} \binom{s_\ell}{f_\ell}^{F_{\ell+1}/F_L} = \prod_{\ell=0}^{L-1} \binom{s_\ell}{f_\ell}^{F_{\ell+1}}$ where $F_L := 1$.

**Stage 2: Distinguishing non-isomorphic trees.** We next show that there exists some parameters $\Theta'$ such that the layer propagation following Eq. 3 generates distinct output for non-isomorphic trees.

*Notational correction to Eq. 3*: In §3.2, we use $\boldsymbol{x}_\ell^i$ to denote the output embedding where $i$ is the *expert* index. This notation is not precise since the same expert can appear as multiple tree nodes, as discussed in the Stage 1 proof above. To make the correction, we instead let $\boldsymbol{x}_\ell^i$ denote the embedding of *node* index $i$[7] for tree-level $L - \ell$, meaning that there can be $\boldsymbol{x}_\ell^i$ and $\boldsymbol{x}_\ell^{i'}$ mapped to the same expert, where $i \neq i'$.

*Including the bias term*: Our proof requires a minor modification of Eq. 3 to add a bias term $\boldsymbol{b}_k^n \in \mathbb{R}^{d_{k+1}}$ associated with each expert $n$. So the updated layer propagation equation, adapted from Eq. 3 now becomes:

$$\boldsymbol{x}_{\ell+1}^i = \sum_{n \in \mathcal{N}_\ell^i} \sigma \left(\boldsymbol{B}_\ell^n \cdot \boldsymbol{A}_\ell^n \cdot \boldsymbol{x} + \boldsymbol{W}_\ell \cdot \boldsymbol{x}_\ell^{i \to n} + \boldsymbol{b}_\ell^n\right) \tag{32}$$

---

[7]In our terminology above, this means that each $(\ell, i)$ now corresponds to a *distinct* ancestral path.

where $\ell$ is the *layer* index; $i$ is the index of a *tree node*, while $n$ is still the index of an *expert*. $\mathcal{N}_\ell^i$ denotes the set of indices of the children experts selected by node $i$. Note, "$i \to n$" means that a tree node $i$ picks a previous-layer expert $n$ as its child. So with a slight abuse of notation, we use superscript "$i \to n$" to index such a child tree node. $\alpha_\ell^{i,n}$ of Eq. 3 is omitted since we simplify the expert weight as binary mask, as stated above.

We are now ready for the proof.

First, note that since the operations by Eq. 3 are permutation invariant, S'MoRE will generate the same output for all isomorphic trees.

Next, we consider non-isomorphic trees. Again we prove by induction.

Similar to the Stage 1 setting, we consider an expert-selection process from layer $\ell$ down to layer 1. After building such an $\ell$-level tree, the model propagates the input token $\boldsymbol{x}$ from layer 1 up to layer $\ell$ to generate the output $\boldsymbol{x}_\ell$. Note, since in the induction step, the propagation terminates at $\boldsymbol{x}_\ell$, we do not need to superscript $\boldsymbol{x}_\ell$ with an additional node index $i$. In other words, $\boldsymbol{x}_\ell$ here is analogous to the *final* embedding $\boldsymbol{x}_L$ described in §3.2.

*Induction hypothesis*: For any $\ell$-level non-isomorphic trees $\mathcal{T}_\ell \neq \mathcal{T}_\ell'$, we can set the layer 1 to $\ell$ parameters of S'MoRE such that $\boldsymbol{x}_\ell \neq \boldsymbol{x}_\ell'$.

*Base case $\ell = 1$*: For a single layer, the propagation simplifies to

$$\boldsymbol{x}_1 = \sum_{n \in \mathcal{N}_0} \sigma \left( \boldsymbol{B}_0^n \cdot \boldsymbol{A}_0^n \cdot \boldsymbol{x} + \boldsymbol{b}_0^n \right) \tag{33}$$

where non-isomorphic trees under $\ell = 0$ degrades to distinct neighbor sets $\mathcal{N}_0$.

We want distinct outputs $\boldsymbol{x}_\ell \neq \boldsymbol{x}_\ell'$ for *all* inputs $\boldsymbol{x}$. So we have the following simple way to construct the parameters:

- $\boldsymbol{B}_0^n = \boldsymbol{0}$ and $\boldsymbol{A}_0^n = \boldsymbol{0}$, which leads to $\boldsymbol{B}_0^n \cdot \boldsymbol{A}_0^n \cdot \boldsymbol{x} + \boldsymbol{b}_0^n = \boldsymbol{b}_0^n$ for all input $\boldsymbol{x}$;

- Let the first element of $\boldsymbol{b}_0^n$ store the expert index (an integer from 1 to $s_0$), and the rest of the elements be 0.

We reuse the following lemma from Xu et al. [2019]:

**Lemma C.8.** *(see Lemma 5 of Xu et al. [2019]) Assume a countable input feature space $\mathcal{X}$. There exists a function $f : \mathcal{X} \to \mathbb{R}^d$ so that $h(X) = \sum_{x \in X} f(x)$ is unique for each set $X \subset \mathcal{X}$ of bounded size.*

In our case, $\sigma$ of Eq. 32 corresponds to function $f$ of Lemma C.8, and we treat $\boldsymbol{b}_0^n$ as the function's input features. The "feature space" consisting of all possible $\boldsymbol{b}_0^n$ is clearly countable (since each element of $\boldsymbol{b}_0^n$ is either 0 or a bounded integer). The neighbor set $\mathcal{N}_0$ corresponds to $X$ of Lemma C.8, which can be an arbitrary combination of the children experts.

Thus, due to the universal approximation theorem [Hornik et al., 1989], we can instantiate $\sigma$ as an MLP to implement such a function $f$, to guarantee that all non-isomorphic trees get a unique output $\boldsymbol{x}_1$. This proves the base case.

*Induction from $\ell$ to $\ell+1$*: Consider two trees constructed by recursive expert selection from layer $\ell+1$ to 1. We use "prime" to denote quantities of the second tree. For example, their leaves' ancestral paths are $\mathcal{T}_{\ell+1}$ and $\mathcal{T}_{\ell+1}'$. According to the analysis in the Stage 1 proof above, there are two possibilities to make the two trees non-isomorphic. i.e., $\mathcal{T}_{\ell+1} \neq \mathcal{T}_{\ell+1}'$:

1. The sets of level-1 nodes are different: $\mathcal{E}_\ell \neq \mathcal{E}_\ell'$;

2. Otherwise, let $\mathcal{E}_\ell = \mathcal{E}_\ell' = \{(\ell+1, i_1), \ldots, (\ell+1, i_{f_\ell})\}$. There exists $i_m$ such that $\mathcal{T}_{\ell, i_m} \neq \mathcal{T}_{\ell, i_m}'$ for some $1 \leq m \leq f_\ell$.

Our goal is to show that for each of the above cases, Eq. 32 can generate distinct outputs for $\mathcal{T}_{\ell+1}$ and $\mathcal{T}_{\ell+1}'$.

Similar to the construction in the $\ell = 1$ case, we set $\boldsymbol{B}_\ell^n = \boldsymbol{0}$ and $\boldsymbol{A}_\ell^n = \boldsymbol{0}$. And $\boldsymbol{b}_\ell^n$ is a one-hot vector with the first element being the expert index (ranging from 1 to $s_\ell$). Recall that $\boldsymbol{W}_\ell \in \mathbb{R}^{d_{\ell+1} \times d_\ell}$ where $d_{\ell+1} = s_\ell \cdot r_\ell + d_\ell$ (see Eq. 4). We set

$$\boldsymbol{W}_\ell = \begin{bmatrix} \boldsymbol{0}_{(s_\ell \cdot r_\ell) \times d_\ell} \\ \boldsymbol{I}_{d_\ell \times d_\ell} \end{bmatrix} \tag{34}$$

where $\boldsymbol{0}_{(s_\ell \cdot r_\ell) \times d_\ell}$ is a $(s_\ell \cdot r_\ell) \times d_\ell$ all-0 matrix and $\boldsymbol{I}_{d_\ell \times d_\ell}$ is a $d_\ell \times d_\ell$ identity matrix.

So Eq. 32 now becomes

$$\boldsymbol{x}_{\ell+1}^i = \sum_{n \in \mathcal{N}_\ell^i} \sigma \left( \begin{bmatrix} \hat{\boldsymbol{b}}_\ell^n \\ \boldsymbol{x}_\ell^{i \to n} \end{bmatrix} \right) \tag{35}$$

where $\hat{\boldsymbol{b}}_\ell^n$ is a length-$(s_\ell \cdot r_\ell)$ vector by discarding the trailing 0s of $\boldsymbol{b}_\ell^n$.

Since the layer-$(\ell+1)$ output corresponds to the tree root, we can ignore the index $i$. Also note that $i \to n$ is essentially $i_m$ of $\mathcal{E}_\ell$ above.

So we have

$$\boldsymbol{x}_{\ell+1} = \sum_{n \in \mathcal{N}_\ell} \sigma \left( \begin{bmatrix} \hat{\boldsymbol{b}}_\ell^n \\ \boldsymbol{x}_\ell^{i_m} \end{bmatrix} \right) \tag{36}$$

Finally, we go back to the two cases above that makes two trees non-isomorphic. Clearly, for either case, the two non-isomorphic trees will have different sets of $\begin{bmatrix} \hat{\boldsymbol{b}}_\ell^n \\ \boldsymbol{x}_\ell^{i_m} \end{bmatrix}$. This allows us to apply Lemma C.8, and conclude that the outputs $\boldsymbol{x}_{\ell+1}$ will also be different for the two non-isomorphic trees.

Note that 1. we are still dealing with a countable feature space, since there are finite number (i.e., $\Gamma_{\texttt{S'MoRE}}^\ell$) of distinct $\boldsymbol{x}_\ell^{i_m}$; 2. Different sets of $\begin{bmatrix} \hat{\boldsymbol{b}}_\ell^n \\ \boldsymbol{x}_\ell^{i_m} \end{bmatrix}$ means different input "$X$" to function $f$ in Lemma C.8.

This completes the induction step from $\ell$ to $\ell+1$.

In sum, our layer propagation function in Eq. 32 ensures that we can find some S'MoRE parameters $\Theta'$ such that all depth-$L$ non-isomorphic trees will lead to distinct outputs $\boldsymbol{x}_L$.

Combining the proof for the two stages, we have shown that the "structural flexibility" of S'MoRE equals

$$\Gamma_{\texttt{S'MoRE}} = \prod_{\ell=0}^{L-1} \binom{s_\ell}{f_\ell}^{F_{\ell+1}}. \tag{37}$$

**Final remark.** In the proof, we require $\sigma$ to be an MLP. In practice, we can implement $\sigma$ simply as non-linear activation (e.g., ReLU). It is easy to see that setting $\sigma$ as "an MLP with a *single* hidden layer of dimension $d_{\ell+1}$" is equivalent to setting $\sigma$ simply as an activation function — For the single-layer MLP, the transformation matrix before the activation can be merged with $\boldsymbol{B}_\ell^n \cdot \boldsymbol{A}_\ell^n$ and $\boldsymbol{W}_\ell$ of the S'MoRE layer. The transformation matrix after the activation can be merged with the next layer $\boldsymbol{W}_{\ell+1}$.

Even if we implement $\sigma$ as an MLP of at least 2 layers, it is still computation and parameter efficient. The input dimension to the MLP is $d_\ell$, which is small (compared with the dimension of the token embeddings). Thus, it is reasonable to set the hidden dimension of the MLP layers also small. This makes the overall MLP very compact. We can follow similar reasoning as §3.4.

$\square$

### C.2.4 Proof of Corollary 3.5

**Corollary C.9.** *(Corollary 3.5) Let $\Gamma^\ell_{S'MoRE*}$ be the structural flexibility of $\ell$-layer S'MoRE variant under Eq. 38. It satisfies the following recursion: $\Gamma^\ell_{S'MoRE*} = \binom{s_{\ell-1}}{f_{\ell-1}} \cdot \binom{\Gamma^{\ell-1}_{S'MoRE*}+f_{\ell-1}-1}{f_{\ell-1}}$, where $\Gamma^0_{S'MoRE*} := 1$.*

*Proof.* This proof utilizes the construction in proving Theorem 3.4.

First, we decompose Eq. 38 as (like before, we ignore router weight $\alpha$ for brevity):

$$x^i_{\ell+1} = \sum_{n \in \mathcal{N}^i_\ell} \left( B^n_\ell \cdot A^n_\ell \cdot x + W_\ell \cdot \sigma\left(x^n_\ell\right)\right) \tag{38}$$

$$= \underbrace{\left(\sum_{n \in \mathcal{N}^i_\ell} B^n_\ell \cdot A^n_\ell \cdot x\right)}_{(a)} + W_\ell \cdot \underbrace{\left(\sum_{n \in \mathcal{N}^i_\ell} \sigma\left(x^n_\ell\right)\right)}_{(b)} \tag{39}$$

We consider how many distinct values (a) and (b) can take.

**Term (b).** Suppose an $\ell$-layer S'MoRE * can generate $\Gamma^\ell_{S'MoRE*}$ distinct outputs, meaning that $x^n_\ell$ can take $\Gamma^\ell_{S'MoRE*}$ different values – This is as if we have a pool of $\Gamma^\ell_{S'MoRE*}$ distinct elements.

The first question is, if we take $f_\ell = \left|\mathcal{N}^i_\ell\right|$ elements from this pool (where the same element can be taken multiple time, since different children $n$ can have the same descendant sub-tree), how many unique multisets[8] can we obtain. This is a classic "combination with replacement" problem, and the solution is $\binom{\Gamma^\ell_{S'MoRE*}+f_\ell-1}{f_\ell}$.

The second question is, can we encode each distinct multiset into distinct outputs via the form of $\sum \sigma(\cdot)$. Reusing Lemma C.8[9], the answer is affirmative.

So term (b) can take $\Gamma^\ell_{S'MoRE*}$ distinct values.

**Term (a).** Since the router takes top-$f_\ell$ experts, there are in total $\binom{s_\ell}{f_\ell}$ distinct $\mathcal{N}^i_\ell$. The key problem is if we perform the simple summation $\sum_{n \in \mathcal{N}^i_\ell}$ without the mapping $\sigma$, can we ensure distinct output for each distinct $\mathcal{N}^i_\ell$ (we cannot apply Lemma C.8 without $\sigma$)? i.e., for any $\mathcal{N}^i_\ell \neq \mathcal{N}^{i'}_\ell$, how can we ensure $\sum_{n \in \mathcal{N}^i_\ell} B^n_\ell \cdot A^n_\ell \cdot x \neq \sum_{n' \in \mathcal{N}^{i'}_\ell} B^{n'}_\ell \cdot A^{n'}_\ell \cdot x$. Setting a bias term encoding the expert index $i$, following Appendix C.2.3, does not work. A failure case is that $\mathcal{N}^i_\ell$ contains experts 1, 4 and $\mathcal{N}^{i'}_\ell$ contains experts 2, 3: $1 + 4 = 2 + 3$ even through $\{1,4\} \neq \{2,3\}$. Fortunately, there are existing encoding schemes that satisfies our requirement. For example, we can encode the $s_\ell$ experts into a "superincreasing sequence" where expert $i$ is encoded into $2^i$. In this case, it is guaranteed that $\sum_{n \in \mathcal{N}^i_\ell} 2^n \neq \sum_{n' \in \mathcal{N}^{i'}_\ell} 2^{n'}$ for any $\mathcal{N}^i_\ell \neq \mathcal{N}^{i'}_\ell$.

**Combining (a) and (b).** Finally, when we set $W_\ell$ according to Eq. 34, we are guaranteed that any two different pairs of (a) and (b) will have different values of "(a) + (b)". This means the total number of distinct $x^i_{\ell+1}$ we can obtain from Eq. 38 equals:

$$\Gamma^{\ell+1}_{S'MoRE*} = \binom{s_\ell}{f_\ell} \cdot \binom{\Gamma^\ell_{S'MoRE*} + f_\ell - 1}{f_\ell} \tag{40}$$

Lastly, when $\ell = 1$, it is obvious that $\Gamma^1_{S'MoRE*}$ should be $\binom{s_0}{f_0}$. If we define $\Gamma^0_{S'MoRE*} := 1$ and let $\ell = 0$, Eq. 40 becomes $\Gamma^1_{S'MoRE*} = \binom{s_0}{f_0} \cdot \binom{1+f_0-1}{f_0} = \binom{s_0}{f_0}$, which satisfies the initial condition.

This completes the proof.

---

[8] A multiset is a set where an element can appear multiple times.

[9] The original Lemma in Xu et al. [2019] is indeed derived on multisets.

$\square$

### C.2.5 Proof of Corollary 3.6

**Corollary C.10.** *(Corollary 3.6) The structural flexibility of* `S'MoRE`# *equals* $\prod_{\ell=0}^{L-1} \binom{s}{f}^{F_{\ell+1}}$ *where* $F_L := 1$.

*Proof.* This proof follows almost exactly as the the proof of Theorem 3.4 in Appendix C.2.3. The only difference is that now every layer has the same dimension $d$, rather than $d$ being increased with larger layer index $\ell$.

This just requires the following minor modification to the proof in Appendix C.2.3:

- When applying Lemma C.8, instead of constructing the mapping $\mathcal{X} \to \mathbb{R}^d$, we instead do the mapping $\mathcal{X} \to \mathbb{R}^{d'}$, with any $d' < d$ (Note that the Lemma does not have constraint on the output dimension $d$). So the output of $\sum \sigma(\cdot)$ is in a $d'$-dimensional subspace of $\mathbb{R}^d$.

- Updating Eq. 34, we set $\boldsymbol{W}_\ell$ to be a projection matrix with the first $d - d'$ rows being 0, and the rest $d'$ rows being a projection from $\mathbb{R}^d$ to the $\mathbb{R}^{d'}$ that $\sum \sigma(\cdot)$ spans.

$\square$

## D  Additional Experimental Results

### D.1  Dataset details

We evaluate on a diverse set of benchmark consisting of 7 popular fine-tuning datasets. Specifically, ARC-c and ARC-e [Clark et al., 2018] evaluate logical reasoning and world knowledge through challenging multiple-choice questions. Commonsense QA [Talmor et al., 2018] assesses a model's grasp of everyday knowledge and implicit relationships. OpenBook QA [Mihaylov et al., 2018] requires multi-step reasoning over scientific facts, while Winogrande [Sakaguchi et al., 2021] measures commonsense pronoun resolution. Accuracy is used as the evaluation metric for all above datasets. In addition, we evaluate the models on 2 more challenging datasets. GSM8K [Cobbe et al., 2021] contains 8.5k high-quality linguistically diverse grade school math word problems. Deriving the correct solution requires multi-step reasoning (2 to 8 steps) by the LLM model. CodeAlpaca [Chaudhary, 2023] contains 20k instruction-following data for fine-tuning LLM's code generation capability. HumanEval [Chen et al., 2021] consists of 164 hand-written programming problems, to access the LLM's capabilities in language comprehension, reasoning, algorithms, and simple mathematics. We train the LLM on CodeAlpaca and then evaluate the checkpoint on HumanEval. We measure the "Pass@1" metric, where we let the fine-tuned model to generate $k = 1$ solution for each problem, and evaluate whether it can pass the unit tests.

### D.2  More details on experimental setup

For all models, we insert the adapters to the feed forward networks (FFN) of all transformer layers of the base models. Specifically, each FFN consists of an "up-projection" matrix, a "gate-projection" matrix and a "down-projection" matrix. We insert the adapter to each of the three matrices.

To ensure a fair comparison, we set an equal budget for trainable adapter parameters and compare different model architecture within this constraint. For LoRA [Hu et al., 2021], we vary the rank $r$ in $\{2^k \mid 0 \le k \le 10\}$, and set the `lora_alpha` parameter as $2 \cdot r$ following standard practice. For MixLoRA [Li et al., 2024a], we adjust the number of experts within $\{4, 8\}$, keep the number of active experts within $\{1, 2, 4\}$[10] (while ensuring that it does not exceed half of the total experts), and the expert dimension within $\{2^k \mid 0 \le k \le 6\}$. For HydraLoRA [Tian et al., 2024], we vary the number of heads in $\{4, 8\}$, and the rank $r$ in $\{2^k \mid 0 \le k \le 8\}$. For `S'MoRE`, in most experiments (except the "scaling-up" study in Table 5), we limit `S'MoRE` to two layers due to resource constraints. We vary the number of experts $(s_0, s_1)$ within $\{(2, 2), (4, 4)\}$: the fanout $(f_0, f_1)$ is $(1, 1)$ when

---

[10]"Number of active experts" is only set for the sparse gates ("noisy top-$k$" and "switch"). For dense gates, the number of active experts equals total number of experts.

Table 7: Wall-clock time (second) comparison

| Method | ARC-c | ARC-e | CSQA | OBQA | Winogrande | Average |
|--------|-------|-------|------|------|------------|---------|
| MixLoRA | 426 | 794 | 3343 | 3539 | 3007 | 2222 |
| S'MoRE | 489 (1.15×) | 957 (1.21×) | 4289 (1.28×) | 4406 (1.24×) | 4014 (1.33×) | 2831 (1.24×) |

$(s_0, s_1) = (2, 2)$ and is $(2, 2)$ when $(s_0, s_1) = (4, 4)$[11]. We vary the expert dimension $(r_0, r_1)$ within $\{(2^k, 2^k) \mid 0 \leq k \leq 6\} \cup \{(2^k, 2^{k+1}) \mid 0 \leq k \leq 5\} \cup \{(2^k, 2^{k+2}) \mid 0 \leq k \leq 4\}$. All baselines and S'MoRE are trained with 2 epochs, with learning rate $1e - 4$. The learning rate follows a cosine schedule.

**Software & hardware.** We implement S'MoRE by adding a customized adapter to the Hugging Face PEFT library [Mangrulkar et al., 2022]. All models are trained via the LLaMA-Factory [Zheng et al., 2024] SFT pipeline, ensuring a consistent execution environment. Similarly, all the evaluations are conducted through OpenCompass [Contributors, 2023b], which is a unified evaluation framework providing a standard API for all considered benchmarks. For the computation hardware, all experiments are run on a single node with 4 NVIDIA A100 80GB GPUs.

### D.3 Wall-clock time & potential system optimizations

While §3.4 ensures that S'MoRE theoretically incurs negligible computation overhead, it is true that without system-level optimization, the multi-layer structure may increase the wall-clock time. Yet, such overhead is small.

**Measurement.** Table 7 shows the wall-clock time to finish training of MixLoRA and S'MoRE, measured on the same machine (with 4 NVIDIA A100 GPUs) and same software environment (based on LLaMA-Factory). The backbone model is LLaMA 3-8B. Trainable parameters of MixLoRA (8 rank-64 experts) and S'MoRE (2 layers, each with 4 rank-64 experts) are comparable.

On average, S'MoRE incurs 24% wall-clock time overhead, which is relatively small. The above measurement is based on S'MoRE under native PyTorch implementation, without any system optimization. It is reasonable to expect that the wall-clock time overhead can be further reduced by applying standard techniques, such as

- CUDA kernel fusion, which combines the operation of multiple S'MoRE layers into a single CUDA kernel. This can effectively reduce the "kernel launch" overhead associated with deeper S'MoRE (in native PyTorch, each layer may require its own "kernel launch").

- Token-level parallelism, which interleaves the processing of different layers across different tokens. This is achievable by custom Triton kernels or `torch.compile(..)` optimization. Such parallelism addresses the load-balance between the router and expert layers (since the router is more lightweight than the expert propagation), which improves GPU utilization. Such parallelism can also break the dependency between the top-down routing and bottom-up propagation, as these two stages can be interleaved across tokens.

### D.4 Routing cost

In Fig. 5, we visualize the router computation cost (Eq. 6) relative to that of the experts' layer propagation (Eq. 3), corresponding to the best-performing models in Table 2. The $x$-axis denotes the different S'MoRE structures (in Table 2, we do not include the results corresponding to the "4-2" and "8-8" S'MoRE architectures, due to space limit). The costs are measured by the total number of arithmetic operations performed by the routers versus by the experts. In general, when the residual rank $r_\ell$ is lower, the cost of routing becomes *relatively* higher (since the router operation is independent of the ranks). However, in all cases, the routing cost is insignificant compared to the cost of expert propagation (at most 26%). This is consistent with our theoretical complexity analysis in §3.4.

---

[11]Same as above, the fanouts are only set for sparse gates. For dense gates, the fanout of layer $\ell$ equals the total number of experts in layer $\ell$

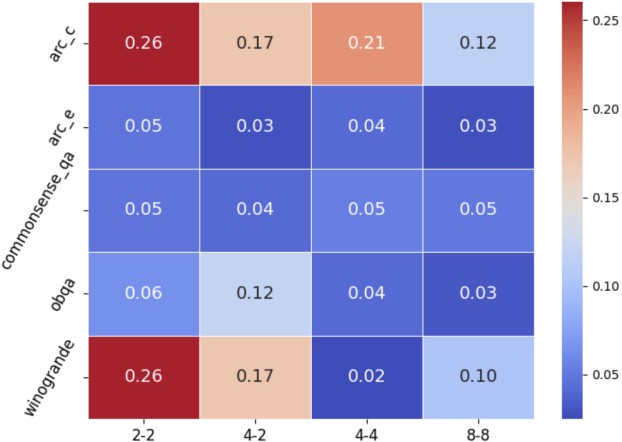

Figure 5: Cost of router (Eq. 6) relative to expert propagation (Eq. 3), measured by their number of arithmetic operations. Here we consider S'MoRE with noisy top-$k$ gate on LLaMA 3.2-1B

# E   Limitations and Broader Impact

**Limitations.** This paper focuses on a novel model architecture design for parameter-efficient MoE, whose computation graph differs from those of standard LoRA and single-layer MoE. We do not focus on the corresponding system-level optimization, and thus our implementation of the S'MoRE layers is written in native PyTorch. To optimally utilize the GPU resources and further accelerate the model execution on commercial hardware, dedicated CUDA kernels may be developed and different levels of execution parallelism (e.g., data-, model- and pipeline-parallelism) may be explored. In addition, we may integrate S'MoRE into state-of-the-art LLM acceleration frameworks such as vLLM [Kwon et al., 2023] or LMDeploy [Contributors, 2023a] to boost the practical execution speed. We treat such system-level optimization as meaningful future work.

**Broader impact.** This work focuses on developing a new PEFT model for the general LLM fine-tuning tasks. It does not have any direct negative societal impact. In the future, S'MoRE may be extended to other tasks or models, to broaden its impact on the enhanced model capacity. For example, we may apply the hierarchical residual design to foundation models under pre-training. The dramatically improved "structural flexibility" under the same parameter and computation budget has the potential to break the ceiling of the current scaling law for both the dense and MoE LLMs.

