# OpenReview forum: "S'MoRE: Structural Mixture of Residual Experts for Parameter-Efficient LLM Fine-tuning"
_NeurIPS.cc/2025/Conference — NeurIPS 2025 poster_

### Official Review · Reviewer_mQH7 · 2025-06-06

**Clarity:** 3
**Significance:** 3
**Originality:** 4
**Rating:** 5
**Confidence:** 4

**Summary:**

This paper proposes S’MoRE (Structural Mixture of Residual Experts), a parameter-efficient fine-tuning (PEFT) framework that integrates the structural capacity of multi-layer MoE with the efficiency of LoRA. The key innovation lies in constructing low-rank expert modules arranged hierarchically, with token-specific routing forming dynamic trees across layers. The authors introduce a theoretical notion of structural flexibility, proving that S'MoRE offers exponentially more routing configurations than flat MoE variants under the same parameter budget. Empirical results across two LLaMA model sizes and seven benchmarks show consistent improvements over strong PEFT baselines.

**Questions:**

Could you include LoRA results with trainable parameter budgets matching those of MixLoRA and S'MoRE (e.g., 0.1B–0.2B)? This would allow a fairer assessment of performance gains attributable to structural design rather than parameter count.

As S’MoRE builds deeper structures with top-down routing, has the runtime or inference latency been evaluated? In practice, does S’MoRE incur significant slowdowns compared to flat MoE or LoRA?

Works like Expert-Specialized Fine-tuning shows that freezing most of the experts could be useful as long as remaining ones are relevant to the task. Is it possible also to train these residual experts on general domains while freezing most of the residual structure and fine-tuning only a subset of experts on a downstream domain? This may potentially further enhance S’MoRE’s modularity.

**Ethical Concerns:**

["NO or VERY MINOR ethics concerns only"]

**Final Justification:**

The authors convincingly showed that S’MoRE outperforms strong PEFT baselines under an identical parameter budget, provided wall-clock results indicating only ~24 % training-time overhead, and clarified that all methods were fairly tuned. I maintain my original score and raise confidence to 4.

**Limitations:**

Yes

**Paper Formatting Concerns:**

No major formatting concerns

**Quality:**

3

**Strengths And Weaknesses:**

**Strengths:**

The work addresses a core challenge in PEFT — scaling capacity under tight budgets — and proposes a novel, theoretically grounded framework that generalizes both LoRA and flat MoE.

Theoretical analysis is rigorous and interpretable. Structural flexibility is formalized with clear upper/lower bounds, and S’MoRE is shown to strictly subsume previous designs.

Viewing routing as dynamic GNN propagation and connecting S’MoRE capacity to graph isomorphism tests adds a novel conceptual layer to PEFT.

The multi-layer structure and recursive routing mechanism are well-illustrated with diagrams and equations.

**Weaknesses:**

In table 2 and 4, LoRA is reported with fewer trainable parameters than the other methods, making the comparison potentially unfair. It would be helpful to include LoRA results matched to the same parameter budget as S’MoRE/MixLoRA.

Despite introducing deeper residual stacks and recursive routing, the paper provides no wall-clock time or latency benchmarks. Since deeper routing trees can serialize otherwise parallel computations, it is important to quantify this tradeoff.

---

> ### Author Rebuttal · Authors · 2025-07-31
>
> We sincerely appreciate the positive feedback and the valuable suggestions!
>
> ---
> > In table 2 and 4, LoRA is reported with fewer trainable parameters than the other methods, making the comparison potentially unfair. It would be helpful to include LoRA results matched to the same parameter budget as S’MoRE/MixLoRA.
>
> > Could you include LoRA results with trainable parameter budgets matching those of MixLoRA and S'MoRE (e.g., 0.1B–0.2B)? This would allow a fairer assessment of performance gains attributable to structural design rather than parameter count.
>
>
>
> Thanks for the question. For Tables 2 and 4, we indeed **enforce the same parameter budget for all methods (including LoRA)**. As described in Appendix D.3, during hyperparameter search, we vary the LoRA’s rank from 1 up to 1024. For S’MoRE, each expert’s rank will not exceed 64. Therefore, the comparison in Tables 2 and 4 are fair.
>
> The parameter count reported in the tables corresponds to the highest-accuracy model in the hyperparameter space (see the table captions). The low parameter count of LoRA indicates that in some cases, LoRA is not able to effectively utilize the available parameters to improve the model capacity. This shows the advantages of the MoE variants (including S’MoRE). In addition, in Figure 4, we plot how the model accuracy scales with increasing parameter count. For the LoRA case (green dots), the accuracy quickly saturates in the low-parameter region.
>
>
> > Despite introducing deeper residual stacks and recursive routing, the paper provides no wall-clock time or latency benchmarks. Since deeper routing trees can serialize otherwise parallel computations, it is important to quantify this tradeoff.
> > As S’MoRE builds deeper structures with top-down routing, has the runtime or inference latency been evaluated? In practice, does S’MoRE incur significant slowdowns compared to flat MoE or LoRA?
>
>
>
> While Sec 3.4 ensures that S’MoRE theoretically incurs negligible computation overhead, it is true that without system-level optimization, the multi-layer structure may increase the wall clock time.
>
> The following table shows the wall clock time to finish training, measured on the same machine (with 4 NVIDIA A100 GPUs). The backbone model is LLaMA 3 8B. Trainable parameters of MixLoRA and S’MoRE are comparable: MixLoRA has 8 rank-64 experts. S’MoRE has 2 layers, each has 4 experts of rank 64.
>
>
> |  |ARC-c|ARC-e|CSQA|OBQA|Winogrande|
> |---|---|---|---|---|---|
> |MixLoRA|426|794|3343|3539|3007|
> |S'MoRE|489 (1.15x) | 957 (1.21x)|4289 (1.28x)|4406 (1.24x)|4014 (1.33x)|
>
> On average, S’MoRE incurs 24% wall-clock time overhead, which is *relatively small*. The above measurement is based on S’MoRE under a native pytorch implementation, without any system optimization. It is reasonable to expect that the wall clock time overhead can be further reduced by applying standard techniques, such as
> * *CUDA kernel fusion*, which combines the operation of multiple S’MoRE layers into a single CUDA kernel. This can effectively reduce the “kernel launch” overhead associated with deeper S’MoRE layers.
> * *Token-level parallelism*, which interleaves the processing of different layers across different tokens, achievable by custom Triton kernels or `torch.compile(..)` optimization. This addresses the load-balance between the router and expert layers (since the router is more lightweight than the expert propagation), which encourages high GPU utilization. Such parallelism design can also break the dependency between the top-down routing and bottom-up propagation, as these two stages of processing can be interleaved across tokens.
>
> However, implementing the above system-level optimization is out of scope of our paper.
>
> ----
> > Works like Expert-Specialized Fine-tuning shows that freezing most of the experts could be useful as long as remaining ones are relevant to the task. Is it possible also to train these residual experts on general domains while freezing most of the residual structure and fine-tuning only a subset of experts on a downstream domain? This may potentially further enhance S’MoRE’s modularity.
>
> This is a great idea and we sincerely appreciate your suggestion! Intuitively, we think of the experts closer to the output layer as the “general experts” that more easily capture the commonality across domains, while the experts closer to the input layer (closer to the leaves in the activated tree) may act more like “specialist” that patch the “general experts” with domain specific structures. Such reasoning is derived from Equation 3: deeper layers have more direct impact on the final output (e.g., for the last layer $L$, $B_{L-1} A_{L-1} x$ is directly fed to the output without the transformation by $W$). So we can understand the last layer experts as the 1st-order residues and the initial layers as higher-order residues.
>
> Following your valuable suggestion, on downstream domains, it may make sense to freeze the deeper layer experts, and just fine-tune the experts in the initial few layers. We are excited about this potential and we totally agree that this further improves the design’s modularity.

---

> > ### Comment · Reviewer_mQH7 · 2025-08-04
> >
> > Thank you for your detailed response. My concerns have been mostly resolved and I will maintain my original score.

---

> > > ### Author Response · Authors · 2025-08-04
> > > **Thank you!**
> > >
> > > Dear reviewer,
> > >
> > > Thank you very much for confirming that most of your concerns have been addressed. We are deeply grateful to your recognition of the novelty, soundness, interpretability and clarify of our work! Your constructive feedback means a lot to us.
> > >
> > > Best,
> > >
> > > Authors

---

### Official Review · Reviewer_zWCQ · 2025-06-26

**Clarity:** 2
**Significance:** 3
**Originality:** 3
**Rating:** 3
**Confidence:** 3

**Summary:**

This paper proposes S’MoRE, a new PEFT method that combines LoRA’s efficiency with MoE’s flexibility by structuring low-rank residual experts into multi-layer trees. It uses a GNN-style router to select expert subtrees per token, enabling high model capacity with minimal parameters. Experiments and theory show it outperforms existing LoRA-MoE hybrids under the same budget.

**Questions:**

1. Line 114: Why can LoRA's BA be viewed as a 1st order approximation. What does 1st order mean here? I don't see how this can be cast into a Taylor approximation type sense (which is what people usually mean). It is a rank r approximation instead. The following sentences seem to imply that this is meant.

2. How to choose the rank of each of the low-rank residuals?

3. What is the difference between r_l and d_L. It becomes confusing when in Line 216 the cost of LoRA and SMORE is compared.

Comments:

1. Can the authors provide a pseudocode to clarify the algorithm?
2. Can the authors include an ablation study isolating the effect of layer depth and tree branching on accuracy/validation metrics

**Ethical Concerns:**

["NO or VERY MINOR ethics concerns only"]

**Final Justification:**

I think this paper has potential, and has a good evaluation.

However, the language to describe the method is ambiguous, see the 1st order approximation concern. The corresponding section needs to be redone, which is why I think the paper should be re-submitted and I keep my score.

**Limitations:**

The paper lacks the address of
- computational overhead in practise

**Quality:**

2

**Strengths And Weaknesses:**

Strengths:

- The authors show that the method generalizes MoLRE and MoMOR
- The idea seems interseting and gives a new angle on MoE and LoRA type PEFT methods

Weaknesses:
- Section 3.2. is very dense and ideas get lost in the notation. (see questions below)
- It’s unclear how sensitive S’MoRE is to hyperparameter choices
- Training cost vs. baseline methods could be better discussed: How does the deeep routing affect wall time cost? The routing mechanism is (intuitively) harder to optimize (on a software level) than "simpler" approaches.

---

> ### Author Rebuttal · Authors · 2025-07-31
>
> # Presentation
>
> > Section 3.2. is very dense and ideas get lost in the notation.
>
> We appreciate the feedback and are fully committed to continuously improving the writing. We have made the following changes:
> * In Sec 3, add a table summarizing all math notations and annotating the parameter’s dimensionality & practical ranges.
> * In Fig 1, mark the names of NN modules (e.g., $A$, $B$), and annotate the input and output dimensions
> * Shrink Sec 3.4 by moving the details on computation cost to Appendix: “Computation cost” can be derived in a similar manner as "parameter efficiency".
>
> ---
> > Line 114: Why can LoRA's BA be viewed as a 1st order approximation …I don't see how this can be cast into a Taylor approximation type sense ... It is a rank r approximation instead.
>
> Correct. By “order”, we don't mean Taylor expansion type of approximation. We simply mean $BA$ is a low-rank approximation, and higher-order terms correspond to ranks of smaller magnitude (e.g., smaller singular values of SVD).
>
> By introducing orders, we allow the model to gradually improve approximation throughout training. This is important since the limited amount of training data in SFT may not allow the model to learn all ranks equally well.
>
> ----
> > Can the authors provide a pseudocode to clarify the algorithm?
>
> Since it's hard to write down pseudocode in the reply box, we walk through an example to clarify the execution details. Consider S’MoRE of 2 layers, each with 4 experts and fanout of 2.
>
> * Given input token $x$, we run top-down routing (Fig 1b; Sec 3.3):
>   * First, downproject $x$ to $x_{down}$ (lines 191 to 194).
>   * For layer 2, generate the score for each of the 4 experts (Eq 6, 7). Since we are at the last layer, input to Eq 6 is just $x_{down}$, without the $k$ terms. Say we select experts 3 and 1.
>   * For the selected expert 3 of layer 2, we next select its 2 children in layer 1: we run Eq 6, 7 again to generate 4 scores. This time, the inputs to Eq 6 include both $x_{down}$ and the key vector $k_1^3$ (subscript 1 denotes the layer 2 parent (we offset layer index by 1); superscript 3 is expert index). Say we select experts 1, 2 of layer 1.
>   * The selected expert 1 of layer 2 runs the same process. Let’s say it selects experts 1, 3 of layer 1 as children.
> * Now we propagate token $x$ along the selected experts bottom-up
>   * In layer 1, experts 1, 2 are aggregated by Eq 3 (for layer 1, $\ell=0$, so the $W_{\ell} \cdot x_\ell^n$ is empty). The output is $x_1^3$ (3 is parent index).
>   * Similarly, expert 1, 3 of layer 1 are also aggregated to generate output $x_1^1$.
>   * In layer 2, experts 3 and 1 are aggregated by Eq 3, by processing token $x$ and children outputs $x_1^3$ and $x_1^1$. The aggregation generates output $x_2$ (the last layer output doesn't have expert index).
> * $x_2$ is up-projected (lines 170-172) as the adapter’s final output.
>
> ---
> # Technical questions
> > What is the difference between $r_\ell$ and $d_L$. It becomes confusing when in Line 216 the cost of LoRA and SMORE is compared.
>
> Please refer to lines 132 to 136 for the following definition:
> * $r_\ell$: rank of each individual expert in layer $\ell+1$ ($\ell$ starts from 0). i.e., $r_\ell$ is the output dimension of $A$ and input dimension of $B$.
> * $d_\ell$: output dimension of each layer $\ell+1$ expert. i.e., $B$ up-projects $r_\ell$ to $d_\ell$.
> * $d_L$: dimension of the final output (i.e., layer $L$’s output). By Eq 4, $d_L$ is the **total rank** of all experts across all layers.
>
> **Why up-projecting $r_\ell$ to $d_\ell$ in each layer?** This is to avoid information loss when aggregating the outputs from different experts (e.g., suppose input token is processed by 4 rank-16 experts. After aggregating the 4 outputs, the sum can span a subspace of up to $4\times 16=64$ dimensions).
> * The “Dimensionality $d_\ell$” paragraph of Sec 3.2 elaborates such reasoning
> * $d_\ell$ is in fact the minimum dimension that allows a multi-layer S’MoRE to exactly express a single-layer counterpart. See the proof of Prop 3.4 in Appendix C.2.1
> * Such up-projection of $d_\ell$ only incurs negligible overhead (Sec 3.4). Yet it brings significant benefits in boosting model capacity (Sec 3.5).
>
> **How does $d_L$ relate to LoRA (line 216)?** from Eq 4, $d_L$ is the total rank for all experts across all layers. Let’s think about the following trajectory of LoRA $\rightarrow$ S'MoRE evolution
>
> * *Step 1*: start from LoRA with rank $d_L$
> * *Step 2* (1-layer MoE): from Step 1, we split $A$ and $B$ into $n$ slices. Each slice of rank $d_L/n$ forms an expert. This 1-layer MoE improves LoRA's model capacity, while keeping the same parameter budget.
> * *Step 3* (S’MoRE): we further arrange the Step-2 experts’ $A$ and $B$ into a multi-layer structure (with additional $W$ fusing the inter-layer information). The multi-layer structure further boosts 1-layer MoE’s model capacity (Theorem 3.4), while still maintaining the parameter efficiency of LoRA.
>
> In other words, LoRA sets the baseline for parameter efficiency. For parameter count, Step 1 $\approx$ Step 2 $\approx$ Step 3. For model capacity, Step 1<Step 2<Step 3, where “<” is strict due to Prop 3.1, 3.2 and Theorem 3.4
>
> ---
> # Experiments
> > How to choose the rank of each of the low-rank residuals?
>
> Theoretically, all $r_\ell$ can be independent hyperparameters.
>
> Practically, due to resource constraints, we only tested the configurations where $r_i \leq r_j$ for $i < j$. For 2-layer S’MoRE, we covered the cases of $r_0=r_1$, $r_0=r_1/2$ or $r_0=r_1/4$ (see Appendix D.3). For 3-layer (Table 4), we tested $r_0=r_1=r_2$, $r_0=r_1=r_2/2$ or $r_0=r_1=r_2/4$.
>
> **Intuition to reduce rank on initial layers**: By Eq 3, deeper layers have more direct impact on the final output (e.g., for the last layer, $B_{L-1} A_{L-1} x$ is directly fed to the output without the $W$ transformation). So we can see the last layer experts as the 1st-order residues and the initial layers as higher-order ones. Naturally, we do not increase rank for higher-order terms.
>
> Specifically, for S’MoRE in Table 2, the rank for each layer is as follows (due to length limit, we only show the LLaMA 3 8B case).
>
> |   |ARC-c|ARC-e|CSQA|OBQA|Winogrande|
> |---|---|---|---|---|---|
> |Dense (2-2)|16-16|16-16|2-2|8-8| 32-32|
> |Dense (2-2)|4-4|8-32|2-2|4-8|8-32|
> |Switch (4-4)|16-16|16-32|1-4|16-16|64-64|
> |Switch (4-4)|8-16|16-16|1-1|4-4|16-16|
>
> In most cases (70%), we can **simply use the same $r_\ell$ for all layers**.
>
> ---
> > Training cost vs. baseline methods
>
> > computational overhead in practise
>
> While Sec 3.4 ensures that S’MoRE theoretically incurs negligible computation overhead, it is true that without system-level optimization, the multi-layer structure may increase the wall clock time. Yet, such overhead is **small**.
>
> **Measurement**: the following table shows the wall clock time to finish training of MixLoRA and S'MoRE, measured on the same machine (with 4 NVIDIA A100 GPUs) and same software environment (based on LLaMA-Factory). The backbone model is LLaMA 3 8B. Trainable parameters of MixLoRA (8 rank-64 experts) and S’MoRE (2 layers, each with 4 rank-64 experts) are comparable.
>
> |  |ARC-c|ARC-e|CSQA|OBQA|Winogrande|
> |---|---|---|---|---|---|
> |MixLoRA|426|794|3343|3539|3007|
> |S'MoRE|489 (1.15x) | 957 (1.21x)|4289 (1.28x)|4406 (1.24x)|4014 (1.33x)|
>
> On average, S’MoRE incurs 24% wall-clock time overhead, which is relatively small. The above measurement is based on S’MoRE under native pytorch implementation, *without any* system optimization. It is reasonable to expect that the wall clock time overhead can be further reduced by applying standard techniques, such as
> * *CUDA kernel fusion*, which combines the operation of multiple S’MoRE layers into a single CUDA kernel. This can effectively reduce the “kernel launch” overhead associated with deeper S’MoRE (in native PyTorch, each layer may require its own "kernel launch").
> * *Token-level parallelism*, which interleaves the processing of different layers across different tokens. This is achievable by custom Triton kernels or `torch.compile(..)` optimization. Such parallelism addresses the load-balance between the router and expert layers (since the router is more lightweight than the expert propagation), which improves GPU utilization. Such parallelism can also break the dependency between the top-down routing and bottom-up propagation, as these two stages can be interleaved across tokens.
>
> However, implementing the above system optimization is **out of scope** of our paper.
>
> ---
> > an ablation study isolating the effect of layer depth and tree branching on accuracy/validation metrics
>
> **Depth**: In Table 4, we have included the ablation on layer depth. We observe that increasing S’MoRE from 2 to 3 layers can further boost accuracy in many cases. Sometimes, parameter count of the 3-layer model can even be less than the 2-layer one.
>
> **Branching**: we apply 2-layer S’MoRE on LLaMA 3 8B, where each S’MoRE layer has 8 experts. We tested 4 different fanout configurations. e.g., 4-2 means that we pick 2 parents in layer 2, and for each selected parent, we further pick 4 children in layer 1.
>
> |Fanout|ARC-c|ARC-e|CSQA|OBQA|Winogrande|
> |---|---|---|---|---|---|
> |2-2|81.36|**91.89**|**81.00**|89.20|84.06|
> |2-4|**82.71**|91.71|80.10|88.80|84.69|
> |4-2|81.36|91.71|80.59|89.40|**85.40**|
> |4-4|81.36|91.71|80.18|**90.00**|84.29|
>
> In general, we do not observe that a specific fanout is always preferable. Intuitively, different fanouts lead to significantly different experts’ structures. Yet the optimal structure is likely **dataset dependent**. Thus, when each layer contains many experts, we should consider tuning the fanout hyperparameter.
>
> Note: for results in the original submission, we did not tune the fanout of S’MoRE. e.g., when each layer contains 4 experts, we simply set all fanouts as 2. For MixLoRA, we vary the number of active experts up to half of the total experts, and take the best result (line 1242, Appendix D.3).

---

### Official Review · Reviewer_NaLk · 2025-07-03

**Clarity:** 2
**Significance:** 3
**Originality:** 3
**Rating:** 5
**Confidence:** 3

**Summary:**

This paper proposes S'MoRE, a novel PEFT framework for large language models. S’MoRE integrates the efficiency of LoRA with the flexibility of MoE by organizing low-rank experts into a multi-layer residual structure. The method routes tokens through hierarchical subtrees of residual experts and aggregates outputs recursively in a graph-inspired architecture. Theoretical analysis shows S'MoRE achieves exponentially greater structural flexibility than traditional MoE under the same parameter budget. Experiments on multiple benchmarks and base models demonstrate consistent performance improvements over baselines under equivalent or modestly higher parameter budgets.

**Questions:**

* Can the residual tree be constructed bottom-up in a single phase instead of routing and aggregation separately?
* Table 1: As the number of layers increases, the overhead becomes non-trivial. Can you discuss the practical implications and whether pruning or sharing strategies might help?
* Table 2: What are the parameter counts during inference for S'MoRE vs. MixLoRA?
* How does S'MoRE scale with larger models? Would the structural gains still justify the increasing overhead?
* The performance gains over MixLoRA are modest in some benchmarks, despite increased parameter count. Can you provide any insights?

**Ethical Concerns:**

["NO or VERY MINOR ethics concerns only"]

**Final Justification:**

* The rebuttal addressed several key concerns, including the feasibility of bottom-up routing, parameter efficiency during inference, and performance trade-offs with increasing depth and expert count.
* The additional results on Gemma2-9B and updated ablations provided useful insights into the scalability and flexibility of the S’MoRE architecture.
* Although some extensions (e.g., large-scale experiments or bottom-up implementation) are left for future work, the authors have demonstrated a strong understanding of the design space and justified their decisions well.
* These clarifications strengthened the submission, and I have adjusted my score accordingly.

**Limitations:**

Yes.

**Paper Formatting Concerns:**

None.

**Quality:**

3

**Strengths And Weaknesses:**

**Strengths**

* **Quality**: The paper provides solid theoretical analysis and comprehensive experiments demonstrating the effectiveness of S'MoRE across benchmarks and routing schemes.
* **Clarity**: The architecture and methodology are generally well explained, with useful figures and clear motivation.
* **Significance**: The proposed structure addresses key challenges in scaling PEFT methods and has potential impact on future adapter designs.
* **Originality**: Introduces a novel hierarchical MoE framework with recursive routing and structural flexibility, extending beyond existing LoRA and MoE approaches.

**Weaknesses**

* The recursive routing and bottom-up aggregation raise questions: could a single-pass bottom-up tree construction suffice? A discussion or ablation of this design choice would strengthen the justification.
* While theoretically efficient, Table 1 shows that computational overhead increases substantially with deeper layers. This may limit practical scalability.
* The method is only tested on Llama 1B and 8B models. How it scales to larger models remains unclear.
* In some cases (e.g., Table 2 on Winogrande), S'MoRE gains are modest over MixLoRA despite having more trainable parameters.
* There is no direct analysis on how performance changes with depth and number of experts per layer. This limits interpretability of the design choices.

---

> ### Author Rebuttal · Authors · 2025-07-31
>
> Thanks for the positive feedback!
>
> ---
> >  could a single-pass bottom-up tree construction suffice?
>
> > Can the residual tree be constructed bottom-up in a single phase instead of routing and aggregation separately?
>
> Great suggestion! Inspired by the reviewer’s thoughts, we propose the following bottom-up router fully compatible with S’MoRE’s structural mixing design.
>
> In bottom-up routing, our goal remains the same as the top-down router: to customize different children experts for different parents. Yet, under bottom-up, the parent index is unknown when we decide the children. So in our design, we will still include a key vector $k$ (analogous to line 186). Yet this $k$ is not directly associated with any specific parent expert. It represents a node position in the routing tree.
>
> To avoid the discussion overwhelmed by notations, we use the following example to illustrate the core idea. Consider a 3-layer S’MoRE, where each layer has a fanout of $f=2$, and has $s=6$ experts. Like our original design, the router still has the following:
>
> * a down-projection matrix that projects the original token $x$ to $d_{down}$-dimensional $x_{down}$ (e.g., $d_{down}=16$).
> * a small MLP in each layer (except layer 1)
>
> In addition, we instantiate a learnable “key” tensor $K$ in each layer. $K$’s last dimension is $d_{down}=16$, and $K$’s leading dimensions depend on the fanout in parent layers (except that for layer 1, $K$’s dimension additionally depends on the number of experts). For example, in layer 1, $K$ has shape (2, 2, 6, 16). In layer 2 and 3, $K$'s shapes are (2, 2, 16) and (2, 16).
>
> The routing proceeds as follows:
> * At layer 1, we compute dot product between $K$ and $x_{down}$ along the last dimension, generating a score tensor of shape (2, 2, 6). Taking the top-2 along the last dimension, we determine the 2 x 2 x 2 children experts for all the 2 x 2 parents.
> * Then we follow Eq 3 to aggregate the children experts' outputs, generating 2 x 2 different output embeddings.
> * In layer 2, we concatenate the 2 x 2 layer-1 output embeddings and the (2, 2, 16)-shaped $K$, along the last dimension. Then we feed the concatenated tensor into layer 2’s router MLP to generate the score distribution over all 6 candidate experts. Taking the top-2 along the last dimension of the MLP output, we now select the 2 x 2 layer-2 experts, conditioned on the layer-1 children.
> * Layer 3 operates similarly.
>
> **Comparison**: From the above, it is clear that the bottom-up router is still computationally efficient (similar to the original top-down router). Both the bottom-up and top-down designs can model the interaction between parent and children experts. The main difference is that the bottom-up router makes routing decisions based on the children’s aggregated embedding, while the top-down router directly consumes the token embedding. So in the bottom-up design, *the gradient can flow back to the lower-layer experts from the router*. This may lead to interesting training behaviors that differ from the top-down case.
>
> Due to limited GPUs, we are unable to run ablation with the bottom-up router. We leave such evaluation as an important future work.
>
> ---
> > While theoretically efficient, Table 1 shows that computational overhead increases substantially with deeper layers. This may limit practical scalability.
>
> > Table 1: As the number of layers increases, the overhead becomes non-trivial. Can you discuss the practical implications and whether pruning or sharing strategies might help?
>
> **Practical depth**: We would like to clarify that practically, we do not expect S’MoRE to have a large depth (e.g., >4):
> * Constructing a 2-layer structure from the SoTA 1-layer MoE is already a **fundamental upgrade**. Such a 2-layer design has rarely been explored in the literature. Yet this new design can already significantly boost model capacity (Theorem 3.4, and Tables 2, 3).
> * Even for a deep 4-layer S’MoRE, the computation overhead is still manageable (no more than 15%). In return, we enjoy much higher model capacity, indicated by the many orders of magnitude increase of the “structural flexibility” (see Fig 2).
>
> **Scalability**: traditionally (before S’MoRE), scaling MoE means increasing expert rank and number of experts, without considering depth – existing designs are just 1-layer variants where *the “depth” dimension does not exist*. S’MoRE expands the scaling horizon by introducing the additional depth dimension. We expect S’MoRE to scale jointly with rank, number of experts and depth.
>
> **Pruning or sharing strategies**: this is a great point and we do believe such strategies can make S'MoRE even more efficient. The computation overhead grows with more layers, because the recursive routing will select many children nodes at the leaf level. Yet, the closer towards the leaf level, the more likely that the same expert may be selected multiple times by different parents. This indicates redundancy, and leads to opportunities of expert merging. Indeed, a similar problem exists in the GNN literature (when a $k$-layer GNN recursively expands $k$-hop neighbors along graph edges, and the same node may be visited many times by different neighbors). We can adapt "node merging" techniques of [1] to our case to eliminate redundancy.
>
> [1] Jia et al. Redundancy-Free Computation for Graph Neural Networks. In KDD 2020.
>
> ---
> > The method is only tested on Llama 1B and 8B models. How it scales to larger models remains unclear.
>
> > How does S'MoRE scale with larger models? Would the structural gains still justify the increasing overhead?
>
> **Results**: Due to very tight GPU resources, evaluating on much larger models (e.g., Gemma 2 27B or LLaMA 3 70B) is impractical for us. However, we have extended our design to Gemma 2 9B. The results on a subset of benchmarks consistently show that S'MoRE significantly improves accuracy with comparable or even better parameter efficiency. We compare MixLoRA of 8 experts and 2-layer S'MoRE (4 experts per layer). Number in each cell denote accuracy and number in parentheses denote parameter count.
>
> | Gemma2-9B |ARC-e|CSQA|Winogrande|
> |---|---|---|---|
> |MixLoRA (8)|83.07 (0.168)|85.83 (0.096)|89.19 (0.315)|
> |S'MoRE (4-4)|**86.60** (0.169)|**86.32** (0.060)|**90.13** (0.315)|
>
> **Community convention**: many SoTA papers published in top-tier conferences (including our baselines, HydraLoRA and MixLoRA)  have also limited their evaluation on base models of similar scales as ours. We believe that our *extension evaluation on 8-9B base models* should sufficiently justify S'MoRE's effectiveness.
>
> **Increasing overhead?**: In fact, on larger base models (with larger token dimension or FFN hidden dimension), our structural composition introduces **smaller overhead**. Following Eq 12, we take the **worst case** of Table 1 and compute its “overhead ratio” under different $d$ (Table 1: $d=4096$). The following table shows that the overhead shrinks with larger base model sizes, making S’MoRE suitable for larger base models.
>
> |L=4|4096|8192|14336 (LLaMA3 8B hidden dim)|
> |---|---|---|---|
> |overhead|15.0%|7.5%|4.3%|
>
> ---
> > In some cases, S'MoRE gains are modest over MixLoRA despite having more trainable parameters.
>
> > The performance gains over MixLoRA are modest in some benchmarks, despite increased parameter count. Can you provide any insights?
>
> Performance on individual benchmarks can be misleading. The rightmost columns of Table 2 summarize the metrics aggregated over all datasets. From the average results, it is clear that S'MoRE achieves significant improvements over all baselines, under both base models and all gate variants.
>
> **Factors affecting performance**: there can be many factors affecting SFT accuracy. e.g., characteristics of the benchmarks, properties of the base model and configuration of the adapters. From our experiences, it is hard to predict the best adapter configuration. e.g., in many cases, increasing rank does not naturally bring accuracy improvements. Such observations apply to all evaluated adapters (including the simple LoRA). This is why we generated Tables 2, 3, 4 after thorough hyperparameter search (see tuning methodology in Appendix D.3). We believe that the conclusion on S’MoRE’s superior performance is trustworthy, despite that in certain cases, other baseline methods still bring values.
>
> ---
> > direct analysis on how performance changes with depth and number of experts per layer.
>
> **Depth**: Table 4 includes ablation on layer depth. We observe that increasing S’MoRE from 2 to 3 layers can further boost accuracy in many cases. In some cases, the 3-layer model may even have lower parameter count.
>
> **Number of experts per layer**: Table 2 and 3 include two S’MoRE configurations: (2-2) and (4-4) for 2 and 4 experts per layer respectively. During rebuttal, we have further increased the number of experts per layer to 8. The following table is based on LLaMA 3 8B. Number in parentheses denote the parameter count.
>
> |Num experts|ARC-c|ARC-e|CSQA|OBQA|Winogrande|
> |---|---|---|---|---|---|
> |2-2|82.37 (0.305)|91.36 (0.090)|81.82 (0.104)|88.20 (0.047)|83.27 (0.190)|
> |4-4|82.37 (0.104)|91.71 (0.305)|82.06 (0.047)|90.00 (0.480)|85.48 (0.714)|
> |8-8|83.05 (0.121)|91.89 (0.473)|81.65 (0.116)|90.00 (0.247)|85.40 (0.247)|
>
> Overall, we do not observe a consistent trend when increasing the number of experts per layer from 4 to 8. The optimal configuration is thus dataset dependent -- different data distribution may correspond to different optimal structure of experts.
>
> ---
> > Table 2: What are the parameter counts during inference for S'MoRE vs. MixLoRA?
>
> We include the parameter counts as follows. Due to space limit, we only include the case for LLaMA 3 8B under "switch" gate.
>
> |  |ARC-c|ARC-e|CSQA|OBQA|Winogrande|
> |---|---|---|---|---|---|
> |MixLoRA (4)|0.066|0.236|0.024|0.239|0.123|
> |MixLoRA (8)|0.012|0.066|0.024 |0.124|0.473|
> |S'MoRE (2-2)|0.067|0.030|0.015 | 0.038|0.123|
> |S'MoRE (4-4)|0.023|0.152|0.024 | 0.152|0.152|

---

> > ### Comment · Reviewer_NaLk · 2025-08-05
> > **Reply to Authors**
> >
> > Thank you for the detailed and thoughtful rebuttal! I appreciate the added analysis on routing strategies, scaling to larger models, and expert configurations. The new results and architectural insights addressed most of my concerns and further highlight the promise of S’MoRE. I have updated my score accordingly.

---

> > > ### Author Response · Authors · 2025-08-05
> > > **Thank you!**
> > >
> > > Dear Reviewer NaLK,
> > >
> > > Thank you very much for confirming that most of your concerns have been addressed. We are deeply grateful to your positive and constructive feedback!
> > >
> > > Best regards,
> > >
> > > Authors

---

### Official Review · Reviewer_Pq8o · 2025-07-03

**Clarity:** 2
**Significance:** 3
**Originality:** 3
**Rating:** 4
**Confidence:** 3

**Summary:**

The paper introduces *S’MoRE (Structural Mixture of Residual Experts)*, a fine-tuning architecture that balances model capacity and parameter efficiency within the Mixture-of-Experts (MoE) framework. Designed as a parameter-efficient fine-tuning (PEFT) method, S’MoRE builds on the idea of exploiting structural dependencies between experts, via applying hierarchical residual decomposition and tree-based routing, to expand representational capacity without a proportional increase in parameters. Comparable in efficiency to Low-Rank Adaptation (LoRA), S’MoRE approximates the functionality of a significantly larger pool of experts while maintaining each residual component as low-rank. The effectiveness of the proposed model is subsequently assessed on a wide range of benchmarks spanning multiple domains.

**Questions:**

- In practice, how should the $d_{\text{down}}$-dimensional representation $x_{\text{down}}$ be selected?

- Additionally, how does this tuning parameter affect the final performance of the model?

- It would be valuable to investigate whether the proposed S’MoRE architecture maintains its superior fine-tuning performance when applied to alternative base models, such as Mistral AI, DeepSeekMoE, or Gemini, rather than relying solely on LLaMA 3.2-1B and LLaMA 3-8B.

**Ethical Concerns:**

["NO or VERY MINOR ethics concerns only"]

**Final Justification:**

I thank the authors for their thoughtful and comprehensive response. The newly added experimental results on Gemma 2 9B, though limited in scope, provide encouraging evidence that the S'MoRE framework can generalize beyond the originally evaluated LLaMA 3 models. The consistency of performance gains with reduced parameter count strengthens the empirical case for the approach and partially mitigates concerns around model generalizability. In addition, while code availability during review is restricted, the authors have provided extensive implementation details, including initialization strategies, hyperparameter configurations, and evaluation procedures, which enhance reproducibility and transparency. Their commitment to open-sourcing the full pipeline upon acceptance is appreciated. The explanation regarding the dimensionality of the routing representation is well-reasoned and supported by ablation studies, which confirm that performance is relatively robust to this choice. This clarifies an important practical aspect of the method. Finally, the authors' attention to detail in correcting reference formatting and improving clarity is appreciated. Given these updates, I find that the original weaknesses around generalizability and implementation clarity have been substantially addressed. While the empirical scope could still be expanded and reproducibility would benefit from immediate code release, I believe the revised submission reflects a stronger case for acceptance. I am therefore increasing my score.

**Limitations:**

Yes.

**Paper Formatting Concerns:**

No significant formatting issues were observed in the manuscript.

**Quality:**

3

**Strengths And Weaknesses:**

**Strengths:**  Experimental evaluations show that S’MoRE achieves improved performance over some baselines and single-layer alternatives, both in terms of predictive accuracy and parameter usage. These results are consistent across two foundation models (LLaMA 3.2 1B and LLaMA 3 8B), 70 diverse fine-tuning benchmarks, various routing strategies, and different scaling regimes.

**Weaknesses:**

- The current study evaluates S’MoRE using only the LLaMA 3.2-1B and LLaMA 3-8B models from [1] as base architectures, which may limit the generalizability of the proposed approach across other model families.

- **Lack of Reproducible Code**: The paper does not include reproducible code for the numerical experimental results, making it difficult to verify the empirical findings. Consequently, my comments are based solely on the results as presented in the paper.

- **Reference format**: The authors should update the publication information for cited works that have already been accepted at conferences or journals. For example, the following preprint appears to have been accepted:

  - Original citation:
    *Edward J Hu, Yelong Shen, Phillip Wallis, Zeyuan Allen-Zhu, Yuanzhi Li, Shean Wang, Lu Wang, and Weizhu Chen. Lora: Low-rank adaptation of large language models. arXiv preprint arXiv:2106.09685, 2021.*

  - Recommended correction:
    *Hu, Edward J., Yelong Shen, Phillip Wallis, Zeyuan Allen-Zhu, Yuanzhi Li, Shean Wang, Lu Wang, and Weizhu Chen. "Lora: Low-rank adaptation of large language models." ICLR 1, no. 2 (2022): 3.*

**References:**

[1]  Grattafiori, Aaron, Abhimanyu Dubey, Abhinav Jauhri, Abhinav Pandey, Abhishek Kadian, Ahmad Al-Dahle, Aiesha Letman et al. "The llama 3 herd of models." arXiv preprint arXiv:2407.21783 (2024).

---

> ### Author Rebuttal · Authors · 2025-07-31
>
> We sincerely appreciate the valuable feedback from the reviewer.
>
> ---
>
> > generalizability of the proposed approach across other model families
>
> >  investigate whether the proposed S’MoRE architecture maintains its superior fine-tuning performance when applied to alternative base models
>
> Thanks for the suggestion. We have extended S'MoRE to the **Gemma 2 9B** base model. Due to the limited GPU resources, we can only evaluate the new experiments on a subset of benchmarks. In the following table, we follow the same experimental setup as described in the paper (see Section 4.1 and Appendix D.2), and we compare the following models:
> * MixLoRA with 8 experts
> * 2 layer S'MoRE with 4 experts in each layer
>
> The number in each cell denote accuracy and the number in parentheses denote the parameter count.
>
> |              | ARC-e         | CSQA          | Winogrande    |
> |--------------|---------------|---------------|---------------|
> | MixLoRA (8)  | 83.07 (0.168) | 85.83 (0.096) | 89.19 (0.315) |
> | S'MoRE (4-4) | **86.60** (0.169) | **86.32** (0.060) | **90.13** (0.315) |
>
> From the above table, it is clear that S'MoRE achieves significant accuracy improvement without increasing parameters. For Commonsense-QA (CSQA), S'MoRE achieves higher accuracy with 37.5% less parameters. The behaviors on Gemma 2 9B are consistent with those on LLaMA 3 1B or 8B.
>
> ------
>
> > Reproducible code
>
> **Full commitment**: We agree with the reviewer on the importance of including reproducible code. As stated in Appendix D.1 ("Open-source statement"), we are *fully committed to open-source the full implementation, including the training and evaluation pipelines*, upon acceptance of the paper.
>
> We would like to provide our code during the rebuttal. Unfortunately, after checking with AC, we are strictly prohibited to provide any link to the reviewer or the AC. At the current stage, there doesn’t seem to be a valid way to provide the code (but we will do so if allowed).
>
> **Additional details**: Despite such constraints, we would like to elaborate our implementation details below. Please let us know if there’s any specific part you’d like to understand more:
> * In Appendix D.3, we have described our experimental setup (including the hardware and software) as well as the hyperparameter search methodology.
> * In addition, for the following hyperparameters, we directly followed the default SFT template in LLaMA-Factory’s official GitHub, and did not tune their values. For all models / datasets, we have
>   * Number of epochs: 2
>   * Gradient accumulation step: 8
>   * Learning rate scheduling: cosine, with warm-up ratio of 0.1
> * To further ease reproducibility, we share some practical tips below:
>   * Initialization strategy significantly affects SFT training stability. For LoRA and MixLoRA, we follow their original setting: each (expert’s) $A$ matrix is initialized by `nn.init.kaiming_uniform_` and each (expert’s) B matrix is initialized to 0. For S’MoRE, the initialization strategy is inspired by LoRA & MixLoRA, and is theoretically backed by Proposition 3.2. **Intuition**:, a multi-layer S’MoRE should be initialized to produce identical output as a single layer MoE. Correspondingly, during the initial stage of training, S’MoRE can quickly recover the capacity of a 1-layer model. Afterwards, it can explore a better structural mixing strategy to go beyond the performance of the corresponding 1-layer model. Therefore, by line 1012 (Appendix C.2.1),
>     * $W_{proj}$ is initialized to all 0: $W_{proj}$ is analogous to concatenation of all $B_i$ of a 1-layer MoE ($i$ is the expert index)
>     * $W_\ell$ and $B_{\ell}^i$ are initialized to be a binary indexing matrix, according to lines 1009 and 1014
>     * $A^i_\ell$ is initialized by `nn.init.kaiming_uniform_`, similar to $A_i$ of 1-layer MoE, or $A$ of LoRA.
> * The evaluation accuracy is highly sensitive to the prompt template. This is the main reason that we evaluate all models under OpenCompass. We reuse the template and other generation parameters (e.g., max generation length, temperature) from OpenCompass' official GitHub repository without any tuning.
> * Some more implementation details: we implement S’MoRE as a new adapter within the huggingface PEFT library. Therefore, inserting S’MoRE adapter into the base model is practically similar to inserting LoRA. Due to version compatibility issues with LLaMA-Factory (version 0.9.2.dev0) and OpenCompass (version 0.3.9), our S’MoRE implementation is on top of PEFT version 0.14.1. In addition to open-source our training and evaluation pipelines in a standalone public repo, we are also working on integrating S’MoRE into the latest PEFT library to further ease community usage.
>
> -----
>
> > In practice, how should the $d_{down}$-dimensional representation $x_{down}$ be selected?
>
> > how does this tuning parameter affect the final performance of the model?
>
> This is a good question. $d_{down}$ is used to down-project the raw token embedding, so that the MLP router of each S’MoRE layer is of negligible computation cost. Thus, as long as we set $d_{down} \ll d$ (where $d$ is the original token dimension, e.g., $d=4096$ or $14336$ for LLaMA), such a $d_{down}$ configuration would be reasonable.
>
> Practically, we can simplify the design choice even further, and just scale $d_{down}$ to be proportional to the number of experts (see reasoning below). In all our experiments, we did not tune $d_{down}$ at all.
>
> The below analogy between the routers of S’MoRE and 1-layer MoE justifies that $d_{down}$ does not need to be large.
> * Router architecture of 1-layer MoE: state-of-the-art 1-layer MoEs (including HydraLoRA and MixLoRA evaluated in our experiments) select experts as follows: the router has a learnable matrix $W_{router}$ of shape $s \times d$, where $s$ is the total number of experts, and $d$ is the token embedding dimension. The router performs matrix-vector multiplication between $W_{router}$ and the input token embedding, to generate an output vector of size $s$. The output vector defines the score for each expert. Then we can select the experts with top-k scores (if using a sparse gate).
> * Router architecture of S’MoRE: consider a simple case that we set $d_{down}$ to be equal to the number of experts. Then we can interpret $W_{down}$ as acting like $W_{router}$ of a 1-layer MoE. In such a setting, S’MoRE’s down-projection step can be seen as an implicit score generation process. Note that S’MoRE includes a lightweight MLP after the down-projection. This MLP is needed only by the multi-layer S’MoRE, to model the conditional probability based on both the input token and the parent experts (see Section 3.3).
>
> Additionally, we run the following ablation experiments, which shows that a small $d_{down}$ suffices. We add 2-layer S’MoRE adapters to the LLaMA3-8B base model, where each S'MoRE layer contains 4 experts.
>
> | $d_{down}$ | ARC-c | CSQA  | OBQA  |
> |--------|-------|-------|-------|
> | 4      | 82.71 | 81.74 | 89.00 |
> | 8      | 84.41 | 80.84 | 88.40 |
> | 16     | 83.05 | 81.33 | 89.60 |
>
> From the above, it is clear that we can practically set $d_{down}$ to be a small value. Increasing $d_{down}$ does not significantly benefit accuracy.
>
> ----
>
> > Reference format
>
> Good suggestion! We have done a careful check over the full reference list and corrected all such references.

---

> > ### Author Response · Authors · 2025-08-06
> > **Gentle reminder & new results**
> >
> > Dear Reviewer Pq8o,
> >
> > **Reminder**: first, we thank you again for your time and efforts in reviewing our paper! Please kindly let us know if there is any remaining question regarding our rebuttal. We are more than happy to discuss further if that would be helpful.
> >
> > **More comprehensive results**: we are now happy to present new results that extend the Gemma2 9B table in our initial rebuttal. In the table below, we have included:
> >
> > * more model architectures: especially variants of MixLoRA (i.e., SoTA 1-layer MoE adapter) with 4 and 8 total experts.
> > * an additional benchmark, HumanEval: now the evaluation on Gemma 2 9B covers 4 representative benchmarks, comprehensively assessing the model's capabilities in multi-step reasoning, world-knowledge, pronoun resolution and code generation.
> >
> > For all models, we set the *same budget* of trainable parameters and perform thorough hyperparameter search (see Appendix D.3). As usual, numbers in parentheses denote the trainable parameters (billion) corresponding to the highest-accuracy model in the hyperparameter space. Again, as promised, when open-sourcing, we will also **release the code related to Gemma 2 9B for reproducing the below results**.
> >
> > |  Gemma 2 9B  | ARC-e         | CSQA          | Winogrande    | HumanEval     | Average           |
> > |---------------|---------------|---------------|---------------|---------------|---------------|
> > | LoRA          | 79.72 (0.289) | 85.91 (0.145) | 87.06 (0.145) | 43.29 (0.072) | 74.00 (0.163) |
> > | MixLoRA (4)   | 85.54 (0.059) | 85.83 (0.096) | 88.79 (0.169) | 43.29 (0.096) | 75.86 (0.105) |
> > | MixLoRA (8)   | 83.07 (0.168) | 85.83 (0.096) | 89.19 (0.315) | 44.51 (0.168) | 75.65 (0.187) |
> > | **S'MoRE** (2-2)  | 86.24 (0.042) | 86.40 (0.169) | 90.13 (0.169) | 44.51 (0.096) | **76.82** (0.119) |
> > | **S'MoRE** (4-4)  | 86.60 (0.169) | 86.32 (0.060) | 90.13 (0.315) | 46.34 (0.060) | **77.35** (0.151) |
> >
> >
> > Consistently with our original rebuttal, S'MoRE consistently achieves *significant accuracy improvement with similar or even less parameters* (see "MixLoRA (4) *vs.* S'MoRE (2-2)", and "MixLoRA (8) *vs.* S'MoRE (4-4)").
> >
> > We hope that our rebuttal as well as the extended results above have provided convincing evidence on the high quality of our design. We appreciate your time and look forward to hearing from you!
> >
> > Best regards,
> >
> > Authors

---

> > > ### Comment · Reviewer_Pq8o · 2025-08-06
> > >
> > > I thank the authors for their thoughtful and comprehensive response. The newly added experimental results on Gemma 2 9B, though limited in scope, provide encouraging evidence that the S'MoRE framework can generalize beyond the originally evaluated LLaMA 3 models. The consistency of performance gains with reduced parameter count strengthens the empirical case for the approach and partially mitigates concerns around model generalizability. In addition, while code availability during review is restricted, the authors have provided extensive implementation details, including initialization strategies, hyperparameter configurations, and evaluation procedures, which enhance reproducibility and transparency. Their commitment to open-sourcing the full pipeline upon acceptance is appreciated. The explanation regarding the dimensionality of the routing representation is well-reasoned and supported by ablation studies, which confirm that performance is relatively robust to this choice. This clarifies an important practical aspect of the method. Finally, the authors' attention to detail in correcting reference formatting and improving clarity is appreciated. Given these updates, I find that the original weaknesses around generalizability and implementation clarity have been substantially addressed. While the empirical scope could still be expanded and reproducibility would benefit from immediate code release, I believe the revised submission reflects a stronger case for acceptance. I am therefore increasing my score.

---

> > > > ### Author Response · Authors · 2025-08-06
> > > > **Thank you!**
> > > >
> > > > Dear Reviewer Pq8o,
> > > >
> > > > We are glad to hear that our responses have substantially addressed your main concerns. We are deeply grateful to your positive and constructive feedback!
> > > >
> > > > Best regards,
> > > >
> > > > Authors

---

### Decision · Program_Chairs · 2025-09-17

**Decision:**

Accept (poster)

**Comment:**

The paper presents a structured low-rank approach within the Mixture-of-Experts (MoE) framework. The proposed method is supported by solid theoretical analysis and a comprehensive set of experiments. The writing is clear and accessible, making it easy to follow the technical details and overall contribution.

Reviewers initially raised concerns about the lack of experiments on large-scale models and requested clarification on certain design choices. These points were well addressed during the rebuttal period. The authors conducted additional experiments on 9B-scale (Gamma 2) models and provided detailed responses to the reviewers’ questions. As a result, all reviewers acknowledged the improvements, and several increased their scores.

Based on my own reading of the paper and the discussion among reviewers, I believe this is a strong and meaningful contribution to the NeurIPS community. I recommend accepting the paper. For the camera-ready version, the authors should include the new results and ensure that all promised changes are incorporated.